# Tendon and motor phenotypes in the *Crtap-/-* mouse model of recessive osteogenesis imperfecta

Matthew William Grol[1†]*, Nele A Haelterman[1], Joohyun Lim[1], Elda M Munivez[1], Marilyn Archer[2], David M Hudson[2], Sara F Tufa[3], Douglas R Keene[3], Kevin Lei[1], Dongsu Park[1], Cole D Kuzawa[4], Catherine G Ambrose[4], David R Eyre[2], Brendan H Lee[1]*

[1]Department of Molecular and Human Genetics, Baylor College of Medicine, Houston, United States; [2]Department of Orthopaedics and Sports Medicine, University of Washington, Seattle, United States; [3]Shriners Hospital for Children, Portland, United States; [4]Department of Orthopaedic Surgery, UT Health Sciences Center, Houston, United States

*For correspondence:
mgrol2@uwo.ca (MWG);
blee@bcm.edu (BHL)

Present address: [†]Department of Physiology and Pharmacology, University of Western Ontario, London, Canada

Competing interests: The authors declare that no competing interests exist.

**Abstract** Osteogenesis imperfecta (OI) is characterized by short stature, skeletal deformities, low bone mass, and motor deficits. A subset of OI patients also present with joint hypermobility; however, the role of tendon dysfunction in OI pathogenesis is largely unknown. Using the *Crtap-/-* mouse model of severe, recessive OI, we found that mutant Achilles and patellar tendons were thinner and weaker with increased collagen cross-links and reduced collagen fibril size at 1- and 4-months compared to wildtype. Patellar tendons from *Crtap-/-* mice also had altered numbers of CD146+CD200+ and CD146-CD200+ progenitor-like cells at skeletal maturity. RNA-seq analysis of Achilles and patellar tendons from 1-month *Crtap-/-* mice revealed dysregulation in matrix and tendon marker gene expression concomitant with predicted alterations in TGF-β, inflammatory, and metabolic signaling. At 4-months, *Crtap-/-* mice showed increased αSMA, MMP2, and phospho-NFκB staining in the patellar tendon consistent with excess matrix remodeling and tissue inflammation. Finally, a series of behavioral tests showed severe motor impairments and reduced grip strength in 4-month *Crtap-/-* mice – a phenotype that correlates with the tendon pathology.

## Introduction

Tendon is a fibrous tissue that connects skeletal muscle to bone to facilitate motion, whereas ligaments connect articulating bones to support joint alignment and function (*Nourissat et al., 2015*; *Screen et al., 2015*). The extracellular matrix of tendons/ligaments is primarily composed of type I collagen as well as smaller quantities of other collagens and proteoglycans (*Kannus, 2000*). During development, the collagen fibrils in tendons and ligaments develop through addition and lengthening before transitioning to the appositional fusion of existing fibers with continued lengthening in postnatal life (*Kalson et al., 2015*). The synthesis and assembly of this collagen-rich matrix are influenced by other minor collagens and proteoglycans as well as by the cross-linking chemistry of type I procollagen fibrils, which in turn regulates fibril size and strength (*Saito and Marumo, 2010*). Like tendon and ligament, the organic matrix of bone consists largely of type I collagen (*Alford et al., 2015*), and disruptions in collagen synthesis and folding are known to negatively impact its biochemical and mechanical properties in connective tissue diseases such as Osteogenesis Imperfecta (OI) (*Lim et al., 2017a*). However, despite evidence of joint mobility phenotypes and motor deficits in OI patients (*Arponen et al., 2014*; *Primorac et al., 2014*), tendon and ligament phenotypes in this disease are relatively understudied.

OI is a heterogeneous group of disorders characterized by variable short stature, skeletal deformities, low bone mass, and increased bone fragility. Approximately 80% of OI cases are caused by dominantly inherited mutations in the genes encoding the α1(I) or α2(I) chains of type I collagen. Mutations in genes responsible for the synthesis, post-translational modification, and processing of collagen, such as cartilage-associated protein (CRTAP), lead to severe, recessive forms of this disease (*Lim et al., 2017a*). In addition to skeletal defects, other connective tissue manifestations, including joint hypermobility and skin hyperlaxity, are observed in a subset of OI patients (*Arponen et al., 2014*; *Primorac et al., 2014*). Our and others' studies have shown that CRTAP forms a complex with Prolyl 3-hydroxylase 1 (*P3h1*) and Cyclophilin B (CypB, encoded by *Ppib*) and is required for prolyl 3-hydroxylation of type I procollagen at Pro986 of chain α1(I) and Pro707 of chain α2(I) (*Hudson and Eyre, 2013*; *Morello et al., 2006*; *Baldridge et al., 2008*). In this regard, loss of either CRTAP or P3H1 leads to loss of this complex and its activity, causing a severe recessive form of OI characterized by short stature and brittle bones (*Morello et al., 2006*; *Barnes et al., 2006*; *Cabral et al., 2007*; *van Dijk et al., 2009*). Collagen isolated from *Crtap*−/− and *P3h1*−/− mice is characterized by lysine over-modifications and abnormal fibril diameter (*Morello et al., 2006*; *Cabral et al., 2007*). While a comprehensive analysis of *Crtap*−/− mice has revealed multiple connective tissue abnormalities, including in bones, lungs, kidneys, and skin (*Baldridge et al., 2010*; *Grafe et al., 2014*), the impact of the loss of CRTAP on tendons and ligaments remains unknown.

Alterations in collagen fibril size and cross-linking have been noted in a limited number of studies using dominant or recessive mouse models of OI (*Chen et al., 2014*; *Terajima et al., 2016*; *Vranka et al., 2010*); however, whether loss of CRTAP impacts tendon and ligament development and structure remains unknown. In this study, we show that *Crtap*−/− mice have weaker and thinner Achilles and patellar tendons at 1 and 4 months-of-age that are hypercellular with a reduction in tendon volume – a phenotype absent at postnatal day 10 (P10). Examining the collagen matrix, we found an increase in stable (irreversible) collagen cross-links at both timepoints, accompanied by alterations in fibril diameter at 4-months compared to wildtype controls. RNA-seq analyses revealed both shared and distinct changes in the transcriptome of the Achilles and patellar tendons of *Crtap*−/− mice compared to wildtype with a predicted activation of transforming growth factor-β (TGF-β) and inflammatory signaling in both tissues. Changes in gene expression assessed by qRT-PCR revealed an upregulation of several tendon markers in *Crtap*−/− Achilles tendons at 1-month, including scleraxis (*Scx*), type I collagen α1 chain (*Col1a1*), lumican (*Lum*), tenascin-C (*Tnc*), and tenomodulin (*Tnmd*). At the same time, markers of vascularization (i.e., CD31), and collagen extracellular matrix (ECM) (i.e., type I collagen α2 chain (*Col1a2*), type II collagen α1 chain (*Col2a1*), type III collagen α1 chain (*Col3a1*), type IX collagen α2 chain (*Col9a2*)) were not different between groups. Many of these changes were also observed in our RNA-seq experiments. Consistent with the gene expression data at 1-month, 4-month-old *Crtap*−/− mice showed increased α smooth muscle actin (αSMA), matrix metalloproteinase-2 (MMP2), and phospho-nuclear factor kappa B (NFκB) in the patellar tendon consistent with excess matrix remodeling and tissue inflammation. These changes in *Crtap*−/− tendons were accompanied by motor deficits and reduced strength at 4 months-of-age. In conclusion, loss of CRTAP in mice causes abnormalities in load-bearing tendons and significant behavioral impairments.

## Results

### Tendon structure, size, and strength are altered in *Crtap*−/− mice at 1- and 4-months

Mice lacking CRTAP present with growth delay, rhizomelia, and severe osteoporosis together with disruption of other connective tissues, including lung and skin (*Morello et al., 2006*; *Baldridge et al., 2010*). To assess abnormalities in the load-bearing tendons from *Crtap*−/− mice, we harvested ankle and knee joints at 1 and 4 months-of-age to histologically examine the Achilles and patellar tendons. At 1-month, *Crtap*−/− mice presented with thinner Achilles and patellar tendons (*Figure 1C,F*) compared to wildtype (*Figure 1A,D*) and heterozygous mice (*Figure 1B,E*) with increased cell density in both structures (*Figure 1M*). By 4 months-of-age, *Crtap*−/− Achilles and patellar tendons remained thinner and hypercellular compared to wildtype and heterozygous mice (*Figure 1G–L,N*). Interestingly, ectopic chondrogenesis was present towards either end of the

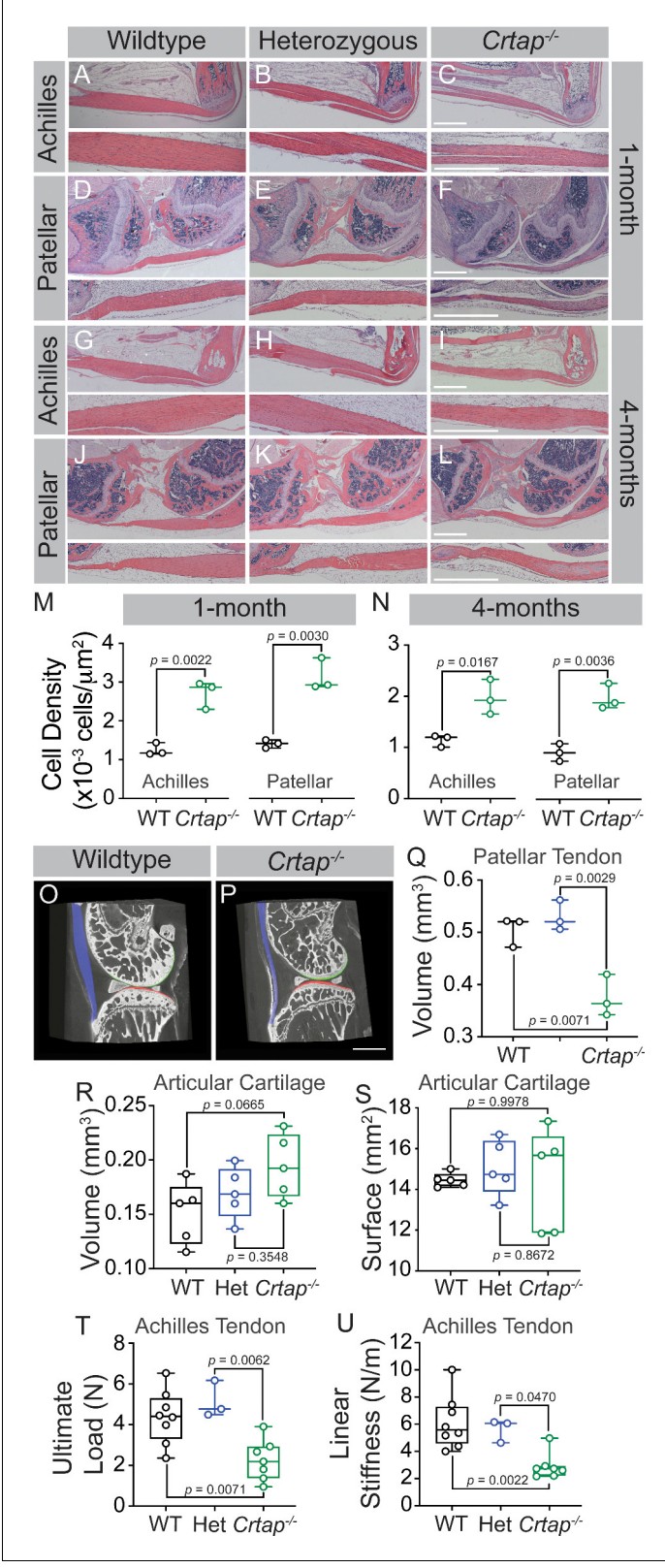

**Figure 1.** Loss of CRTAP causes thinning, hypercellularity, and weakening of tendons in young and mature mice. (A–C) Representative H & E images of 1-month ankle joints. (D–F) Representative H & E images of 1-month knee joints. (G–I) Representative H & E images of 4-month ankle joints. (J–L) Representative H & E images of 4-month knee joints. For all micrographs, higher magnification images of the mid-tendon are illustrated. n = 3–4 mice per

*Figure 1 continued*

group. Scale bar is 1 mm. (**M–N**) Quantification of cell density for Achilles and patellar tendons of wildtype and *Crtap*[-/-] mice at 1 month (**M**) and 4 months (**N**) of age. Data are min-to-max box and whisker plots with individual points indicated. n = 3 mice per group. Data passed the Shapiro-Wilk test for normality, and groups were compared using two-tailed unpaired t-tests. Exact *p*-values are reported. (**O–P**) Representative phase-contrast µCT images of 4-month wildtype (**O**) and *Crtap*[-/-] (**P**) knee joints. Blue indicates the patellar tendon, green indicates the femoral articular cartilage, and red indicates the tibial articular cartilage. Scale bar is 1 mm. (**Q**) Quantification of the patellar tendon volume in wildtype, heterozygous, and *Crtap*[-/-] mice at 4-months. Data are min-to-max box and whisker plots with individual points indicated. n = 3 mice per group. Data passed the Shapiro-Wilk test for normality, and groups were compared using one-way ANOVA with Tukey's post-hoc tests. Exact p-values are reported. (**R–S**) Quantification of articular cartilage volume (**R**) and surface (**S**) in wildtype, heterozygous, and *Crtap*[-/-] mice at 4-months. Data are min-to-max box and whisker plots with individual points indicated. n = 5 mice per group. Data passed the Shapiro-Wilk test for normality, and groups were compared using one-way ANOVA with Tukey's post-hoc tests. Exact p-values are reported. (**T–U**) Biomechanical assessment of ultimate load (**T**) and linear stiffness (**U**) for Achilles tendons from 1-month-old wildtype, heterozygous, and *Crtap*[-/-] mice. Data are min-to-max box and whisker plots with individual points indicated. n = 3–8 mice per group. Data passed the Shapiro-Wilk test for normality, and groups were compared using one-way ANOVA with Tukey's post-hoc tests. Exact p-values are reported.

patellar tendon in some (but not all) 4-month-old *Crtap*[-/-] mice (*Figure 1L*) – a phenomenon that can occur in tendinopathy (*Steinmann et al., 2020*). Consistent with our histological data, phase-contrast µCT analysis demonstrated that patellar tendon volume was reduced in *Crtap*[-/-], but not in heterozygous mice, compared to wildtype controls (*Figure 1O–Q*). In contrast, no significant changes in articular cartilage volume or surface were observed (*Figure 1R,S*).

To determine whether the altered structure of *Crtap*[-/-] load-bearing tendons results in reduced tissue strength, we performed biomechanical testing at 1-month on Achilles tendons to examine structural properties. As predicted based on the histological data, we observed decreases in ultimate load and linear stiffness for *Crtap*[-/-] Achilles tendons compared to heterozygous and wildtype mice (*Figure 1T,U*). Taken together, load-bearing tendons in *Crtap*[-/-] mice present with reduced size, increased cell density and decreased strength compared to controls.

Given *Crtap* is deleted throughout development in our global knockout mouse model, we next examined changes in the Achilles and patellar tendons at postnatal day 10. Interestingly, the *Crtap*[-/-] Achilles tendons (*Figure 2A,B*) but not the patellar tendons (*Figure 2C,D*) were slightly thinner (although not significant) compared to wildtype mice (*Figure 2F*), with no differences in cell density seen in either tissue (*Figure 2E*). Taken together, this data suggests that the tendon phenotypes observed at 1-month and beyond occur postnatally and are not due to defects in tendon development.

Given the significant hypercellularity seen in the load-bearing tendons from *Crtap*[-/-] mice, we next examined whether there were changes in tenocyte populations associated with progenitors and tendon repair response in *Crtap*[-/-] mice. Specifically, previous literature has demonstrated that progenitor-like cells involved in tendon maturation and repair are marked by the expression of CD146 (*Lee et al., 2015*) in addition to others. Using fluorescence-activated cell sorting (FACS) analysis of 5-month-old patellar tendons, we observed a significant decrease in the percentage of CD45[-]CD31[-]CD146[+]CD200[+] (~2%) compared to wildtype mice (~4%) (*Figure 3A–C*, red box). In contrast, CD45[-]CD31[-]CD146[-]CD200[+] cells were concomitantly increased in *Crtap*[-/-] mice (*Figure 3D*). Taken together, this data suggests that the matrix disruptions caused by loss of CRTAP may lead to the dysregulation of discrete tendon cell populations within the adult patellar tendon.

## Collagen fibril formation is altered in heterozygous and *Crtap*[-/-] mice

Tendons develop embryonically by increasing in fibril length and number, whereas postnatal growth arises from an increase in fibril length and diameter – the latter of which is driven by the lateral fusion of smaller fibrils (*Kalson et al., 2015*). To investigate the role of CRTAP in postnatal collagen fibril maturation, we utilized transmission electron microscopy (TEM) to examine changes in fibril diameter in flexor digitorum longus (FDL), Achilles, and patellar tendons (*Figure 4*). In the FDL tendon, there was a marked increase in small collagen fibrils (20–60 nm in size), a reduction in 80–320 nm fibrils, and a slight increase in larger fibrils (>340 nm in diameter) in *Crtap*[-/-] mice compared to

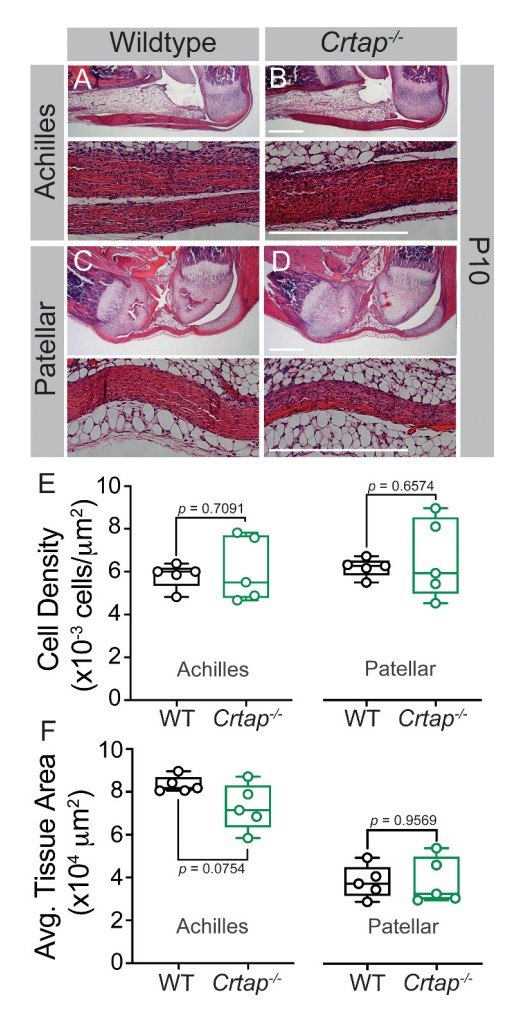

**Figure 2.** Tendon thinning and hypercellularity are not observed in *Crtap*[-/-] mice at postnatal day 10. (**A–B**) Representative H & E images of postnatal day 10 (P10) ankle joints. (**C–D**) Representative H & E images of P10 knee joints. For all micrographs, higher magnification images of the mid-tendon are illustrated. n = 5 mice per group. Scale bar is 0.5 mm. (**E**) Quantification of cell density and (**F**) Average tissue area for Achilles and patellar tendons of wildtype and *Crtap*[-/-] mice at P10 (taken mid-tendon). Data are min-to-max box and whisker plots with individual points indicated. n = 5 mice per group. Data passed the Shapiro-Wilk test for normality, and groups were compared using two-tailed unpaired t-tests. Exact p-values are reported.

wildtype (*Figure 4A,C,J*). Despite similarities seen in histology, heterozygous mutant FDL tendons also exhibited a slight increase in 20–40 nm fibrils in mice compared to wildtype controls (*Figure 4A–B,J*). Similar trends were observed for the Achilles tendon, namely an increase in small fibrils (20–60 nm), a reduction in 80–240 nm fibrils, and an increase in large fibrils (>280 nm) upon loss of *Crtap* (*Figure 4D,F,K*). In contrast to what we observed for the FDL, heterozygous Achilles tendons did not have increased numbers of smaller fibers (*Figure 4E,K*). Instead, a greater number of fibrils ranging from 140-to-200 nm in size were noted compared to wildtype controls.

Compared to the FDL and Achilles tendons, the most significant differences were seen within the patellar tendon, although the pattern of changes remained consistent (*Figure 4G–I,L*). Specifically, we observed a dramatic increase in 20 nm collagen fibrils compared to heterozygous and wildtype mice (*Figure 4G–I,L*). Fibrils ranging from 100-to-180 nm in diameter were reduced in heterozygous and *Crtap*[-/-] mice compared to wildtype. Interestingly, the greatest difference from wildtype was an increase in large collagen fibrils (>200 nm) in both heterozygous and *Crtap*[-/-] mice (*Figure 4G–I,L*).

To examine how collagen fibril alignment is affected by the loss of CRTAP, we examined longitudinal sections of FDL, Achilles, and patellar tendons from 4-month wildtype and *Crtap*[-/-] mice using TEM (*Figure 4M–R*). Consistent with our transverse data, we observed a wider array of thinner and thicker collagen fibrils in *Crtap*[-/-] tendons (*Figure 4P,Q,R*) compared to wildtype (*Figure 4M,N,O*). In addition, while the collagen fibrils in wildtype animals were well-aligned, collagen fibril alignment in *Crtap*[-/-] tendons was more irregular (*Figure 4P,Q,R*). Taken together, these data indicate that loss of CRTAP alters collagen fibril assembly and alignment in load-bearing tendons. In addition, the degree to which collagen assembly is affected is site-dependent.

## Collagen cross-linking is increased in heterozygous and *Crtap*[-/-] mice

Along with P3H1 and CyPB, CRTAP is an integral part of the collagen prolyl 3-hydroxylation complex responsible for the 3-hydroxylation of Pro986 of the type I procollagen α1 chain (*Lim et al., 2017a*). Loss of this complex blocks 3-hydroxyproline formation and affects lysine hydroxylation and cross-linking in bone collagen (*Morello et al., 2006*; *Baldridge et al., 2008*); however, whether *Crtap*[-/-] tendons display altered collagen cross-linking is unknown. To investigate this, we harvested tendons at 1- and 4-months and assessed collagen cross-linking by quantifying the levels of hydroxylysyl-pyridinoline (HP) (*Figure 5*). Overall, we observed an increase in these stable, mature collagen cross-links from 1 to 4 months-of-age in all genotypes for the FDL and Achilles tendons (*Figure 5A–B*). In contrast, for the patellar tendon, age-

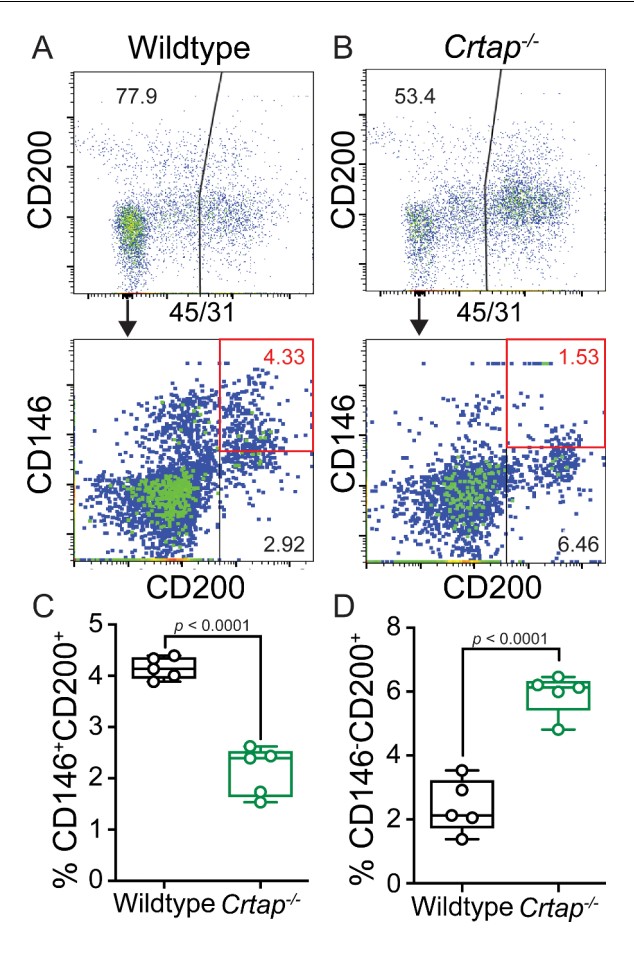

**Figure 3.** Loss of CRTAP in the patellar tendon leads to a decrease in progenitor cells and an accumulation of immature resident tissue cells. (A–B) Patellar tendon cells isolated from 5-month-old wildtype (A) or *Crtap*[-/-] (B) mice were analyzed for the expression of CD200 and CD146 tendon progenitor markers (*top histogram*) within the CD45[-]CD31[-] population (*bottom histogram*). The plots are representative from a single wildtype or *Crtap*[-/-] mouse. (C–D) Graphs show the percentage of CD45[-]CD31[-]CD146[+]CD200[+] progenitor cells (C) and CD45[-]CD31[-]CD146[-]CD200[+] immature tendon cells (D) From 5-month wildtype and *Crtap*[-/-] patellar tendons. Data are min-to-max box and whisker plots with individual points indicated. n = 5 mice per group. Data passed the Shapiro-Wilk test for normality, and groups were compared using two-tailed unpaired t-tests. Exact p-values are reported.

dependent increases in collagen cross-links were only observed in *Crtap*[-/-] mice (*Figure 5C*). For FDL tendons, *Crtap*[-/-] mice had more of these collagen cross-links at 1- and 4-months compared to heterozygous and wildtype mice; however, the content of HP residues per collagen decreased with age in this tissue (*Figure 5A*). Interestingly, in Achilles tendons, an increase in collagen cross-linking was observed in both heterozygous and *Crtap*[-/-] mice at 1-month compared to wildtype. In contrast, at 4-months, only *Crtap*[-/-] mice had elevated collagen cross-links, and these levels were greater than those observed at the earlier time point (*Figure 5B*).

The patellar tendon showed the greatest increase in collagen cross-links both with time and across genotypes of the tissues examined. Specifically, collagen cross-links were elevated by 5- to 10-fold in *Crtap*[-/-] patellar tendons compared to heterozygous and wildtype at 1- and 4-months, respectively (*Figure 5C*). Taken together, these data suggest that CRTAP is required for proper hydroxylation and cross-linking of collagen fibrils in tendons in a semi-dominant fashion, as heterozygous mutant tendons display a phenotype that is milder than the phenotype observed for homozygous mutant mice. Notably, the chemical quality of collagen cross-linking appears to be

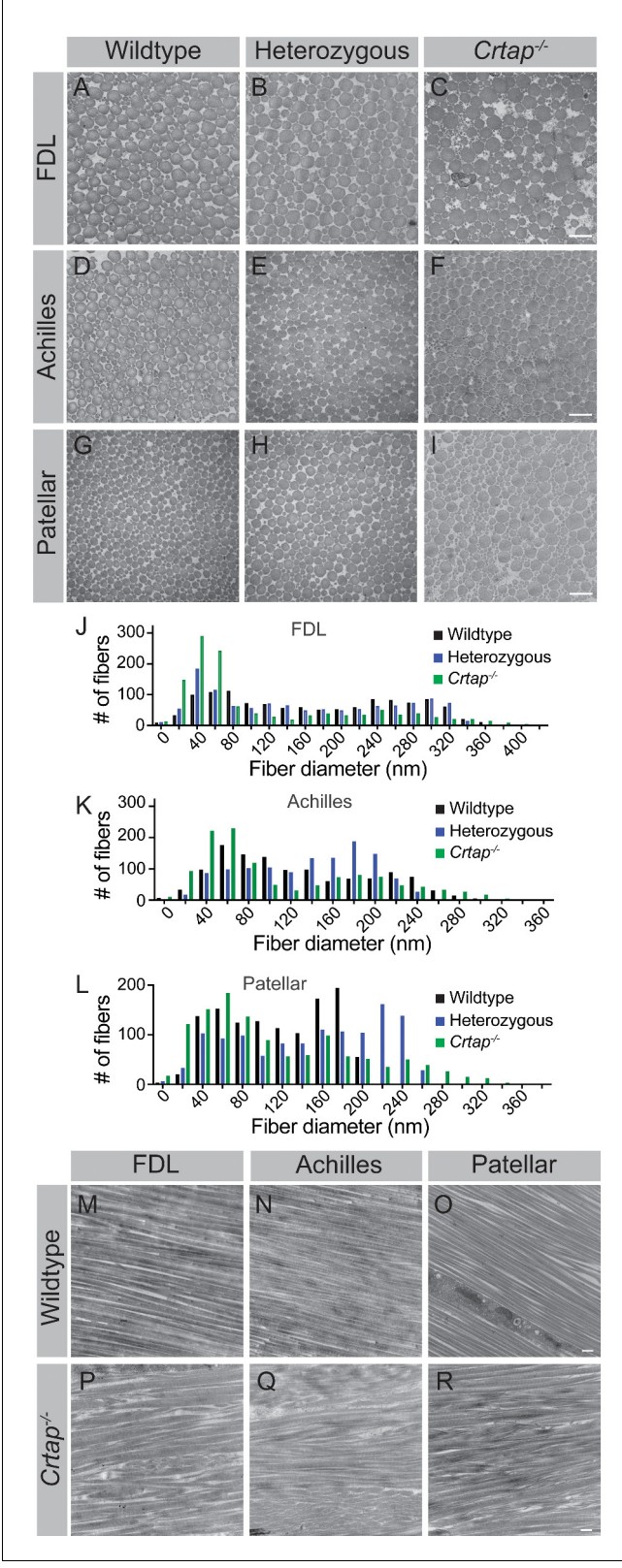

**Figure 4.** Collagen fibril diameter is altered in tendons from heterozygous and *Crtap*⁻/⁻ mice. (A–C) Representative transverse TEM images of 4-month (A–C) FDL tendon collagen fibrils, (D–F) Achilles tendon collagen fibrils, and (G–I) patellar tendon collagen fibrils. Scale bar is 500 nm. (J) Representative histogram of the size distribution for collagen fibrils in FDL tendons. Data are representative of n = 3 mice. (K) Representative histogram of the size

*Figure 4 continued on next page*

*Figure 4 continued*

distribution for collagen fibrils in Achilles tendons. Data are representative of n = 3 mice. (**L**) Representative histogram of the size distribution for collagen fibrils in patellar tendons. Data are representative of n = 3 mice per group. (**M–R**) Representative longitudinal TEM images of 4-month wildtype FDL (**M**), Achilles (**N**), and patellar (**O**) tendons, and 4-month *Crtap*$^{-/-}$ FDL (**P**), Achilles (**Q**), and patellar (**R**) tendons. Scale bar is 500 nm.

spatiotemporally regulated, and this regulation is differentially affected by the loss of a single or both copies of *Crtap*.

## Signaling and metabolic dysregulation in *Crtap*$^{-/-}$ load-bearing tendons

We performed bulk RNA-seq with RNA isolated from Achilles and patellar tendons of 1-month-old wildtype and *Crtap*$^{-/-}$ mice to investigate the molecular changes underlying the observed tendon phenotypes. To determine global changes in differentially expressed genes and predicted upstream regulators, we performed Ingenuity Pathway Analysis (IPA, Qiagen, Germany). For the Achilles tendon, a total of 178 genes (consisting of 99 upregulated genes and 79 downregulated genes) were significantly differentially expressed between wildtype and *Crtap*$^{-/-}$ samples (**Figure 6A**). Of the top 30 differentially expressed genes, several ECM proteins, including matrilin-3 (*Matn3*), matrilin-4 (*Matn4*), and fibronectin 1 (*Fn1*), and proteolytic enzymes such as matrix metallopeptidas-2 (*Mmp2*) were dysregulated. Gene ontology analysis revealed 'GO:000715 – Cell Adhesion', 'GO:0045778 – Positive Regulation of Ossification', and 'GO:0051928 – Positive Regulation of Calcium Ion Transport' to be enriched (**Figure 6B**). Examination of upstream regulators based on the differential gene expression data revealed a predicted activation of TGF-β1 in *Crtap*$^{-/-}$ mice and predicted inhibition of dystrophin (DMD) along with several for which activation state was unclear, including platelet-derived growth factor-BB (PDGF-BB), β-catenin (CTNNB1), and tumor necrosis factor (TNF) (**Figure 6C**).

In keeping with the increased phenotypic severity seen in patellar tendons from *Crtap*$^{-/-}$ mice, a greater number of total genes were differentially expressed between wildtype and *Crtap*$^{-/-}$ patellar tendons (**Figure 6D**). We saw significant differential expression of 519 genes, with 224 being upregulated and 295 downregulated in *Crtap*$^{-/-}$ compared to wildtype. Several of the top 30 differentially expressed genes were minor collagens such as type IX collagen α3 chain (*Col9a3*), type IX collagen α1 chain (*Col9a1*), and type XXII collagen α1 chain (*Col22a1*), as well as other ECM proteins, including lumican (*Lum*) and fibronectin 1 (*Fn1*). Unlike the Achilles tendon, gene ontology analysis revealed significant enrichment for metabolic processes, including 'GO:0055114 – Oxidation-Reduction Process', 'GO:0050873 – Brown Fat Cell Differentiation', and 'GO:0050872 – White Fat Cell Differentiation' as well as for 'GO:0008284 – Positive Regulation of Proliferation' and 'GO:0001525 – Angiogenesis' (**Figure 6E**). Examination of upstream regulators also suggested a highly significant activation of TNF and interleukin 6 (IL-6) as well as fibroblast growth factor 2 (FGF2) and TGF-β3 in *Crtap*$^{-/-}$ patellar tendons (**Figure 6F**). Indeed, the representation of TNF, TGF-β, and PDGF-BB in this dataset is consistent with but more severe than that observed for those same regulators in the Achilles tendon results (**Figure 6C**).

We next examined the expression patterns of genes that are implicated in postnatal tendon maturation and homeostasis. We confirmed the loss of *Crtap* expression in *Crtap*$^{-/-}$ Achilles tendon compared to wildtype – a finding also demonstrated in our RNA-seq data (RNA-seq data shown in red as 'log$_2$ fold change; adjusted p-value' with the graphed qPCR results in **Figure 7**). Interestingly, loss of CRTAP led to a modest upregulation in *Scx* expression with no changes in *Mkx* and *Egr1* transcript levels (**Figure 7**). Despite a modest downregulation in *Col1a1* transcripts in *Crtap*$^{-/-}$ Achilles tendons, there were no changes in the expression of *Col1a2*, *Col2a1*, *Col3a1*, or *Col9a2*. On the other hand, we observed an increase in expression of several ECM glycoproteins and small leucine-rich proteoglycans (SLRPs), including *Tnc*, *Lum*, *Tnmd*, and *Fn1*, known to be upregulated during tendon maturation and repair. We also observed an upregulation in *Thbs3* transcripts in *Crtap*$^{-/-}$ Achilles tendons compared to wildtype, consistent with our RNA-seq data (**Figure 7**). Overall, we speculate that the modest reduction in *Col1a1* expression and altered expression of *Scx*, as well as other ECM proteins such as *Tnc*, *Lum*, and *Tnmd*, could be associated with the observed aberrations in collagen fibrillogenesis seen in *Crtap*$^{-/-}$ mice, and may indicate an increased remodeling or repair response similar to that seen in OI bone (**Grafe et al., 2014**).

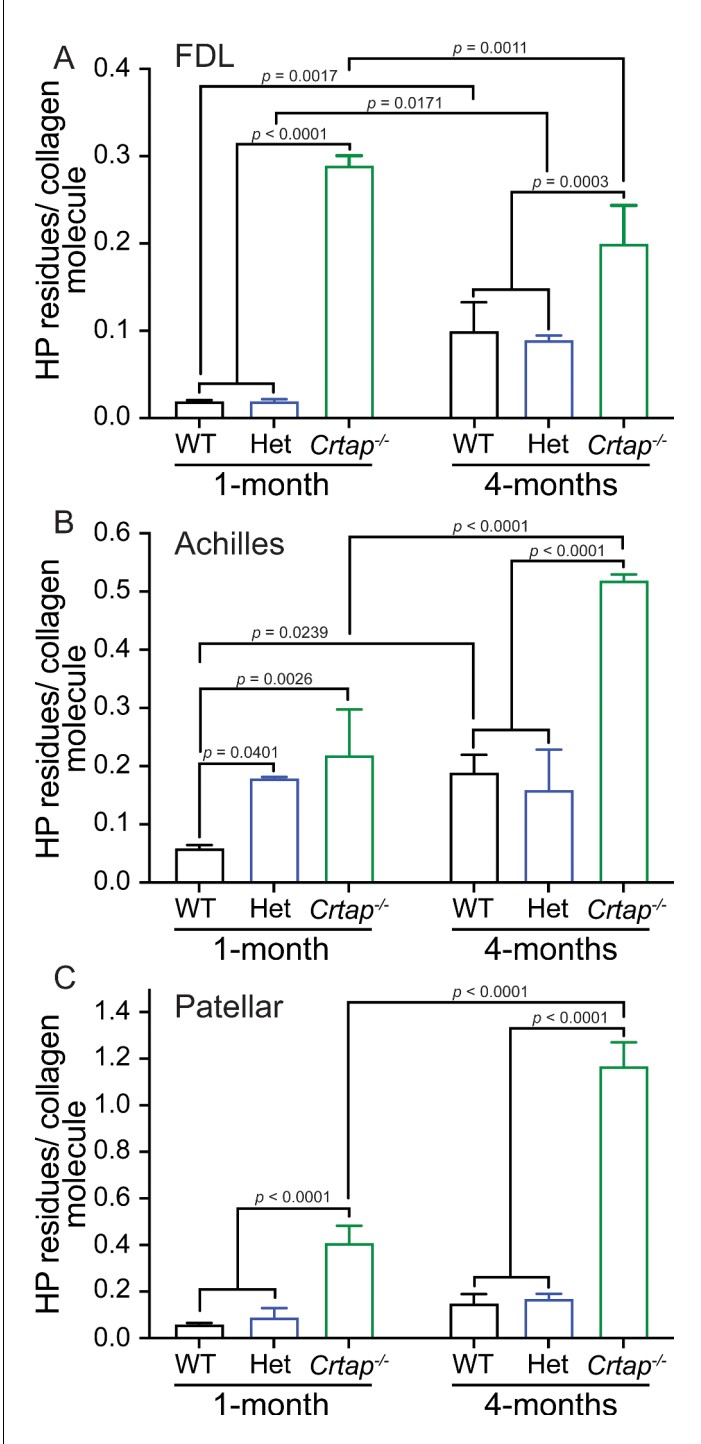

**Figure 5.** Collagen cross-linking is increased in tendons from young and mature *Crtap*−/− mice. Quantification of collagen cross-links as hydroxylysyl-pyridinoline (HP) residues per collagen molecule for (**A**) FDL tendons; (**B**) Achilles tendons; and (**C**) patellar tendons. Data are shown as means ± S.D. n = 3–4 mice per group. Data passed the Shapiro-Wilk test for normality, and groups were compared using one-way ANOVA with Tukey's post-hoc tests. Exact p-values are reported.

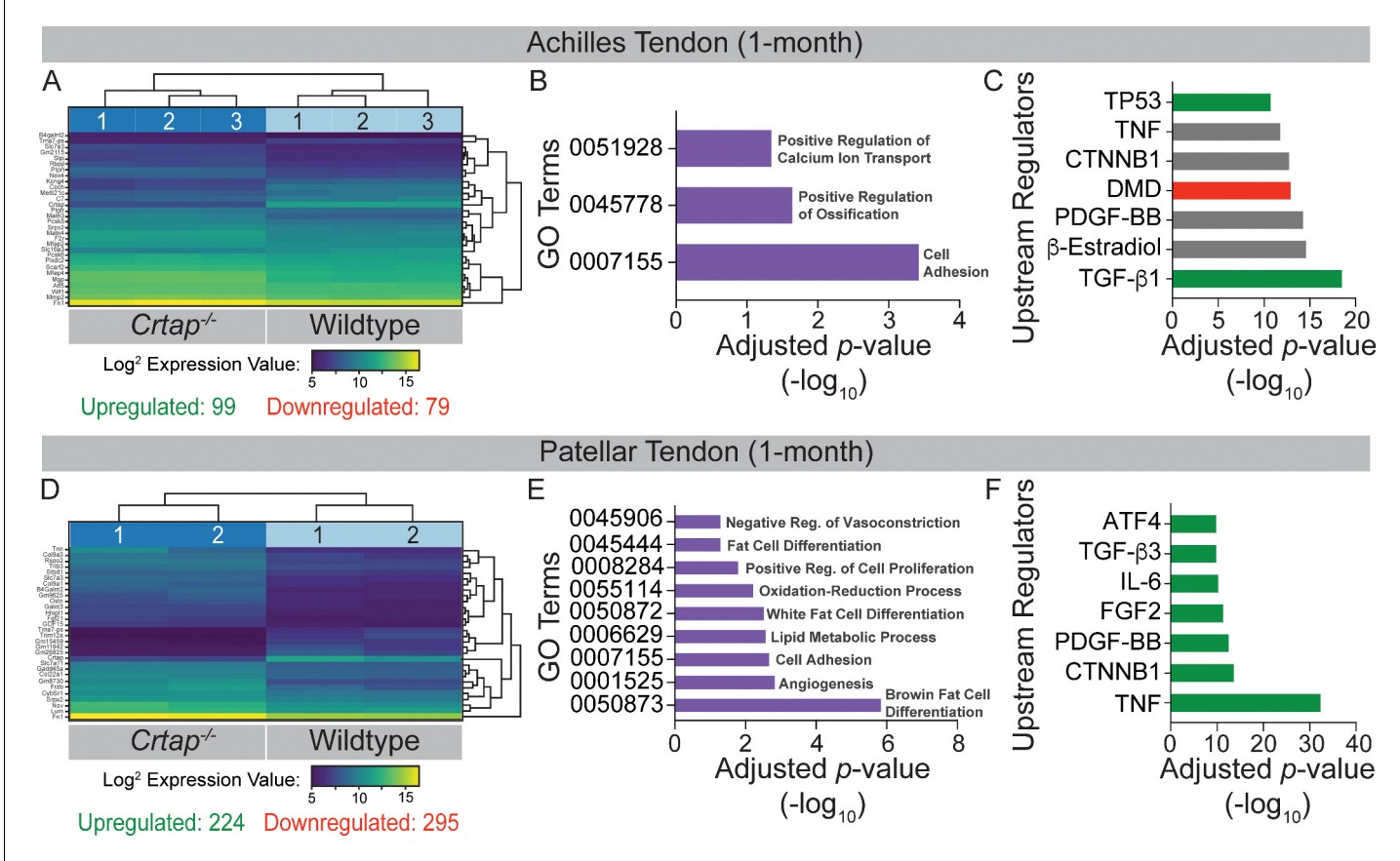

**Figure 6.** Transcriptome analysis of tendons from 1-month-old *Crtap*⁻/⁻ mice. DESeq2 was used to compare gene expression between wildtype and *Crtap*⁻/⁻ Achilles and patellar tendon RNA samples, and genes with an adjusted p-value<0.05 and absolute $\log_2$ fold change >1 were considered as differentially expressed. (**A**) A bi-clustering heatmap of the top 30 differentially expressed genes between wildtype and *Crtap*⁻/⁻ Achilles tendons sorted by adjusted p-value and plotted according to $\log_2$ transformed expression values. The Wald test was used to generate p-values and $\log_2$ fold changes. See *Figure 6—source data 1* for a complete list of differentially regulated genes. (**B**) Significantly differentially expressed genes between Achilles tendons from wildtype and *Crtap*⁻/⁻ mice were clustered by their gene ontology, and enrichment for gene ontology terms was tested using Fisher exact test. All gene ontology terms with an adjusted p-value<0.05 are plotted according to their $-\log_{10}$ adjusted p-value. (**C**) Select upstream regulators predicted as being activated (shown in green), inhibited (shown in red), or of unclear state (shown in gray) in *Crtap*⁻/⁻ compared to wildtype Achilles tendons plotted according to their $-\log_{10}$ adjusted p-value. See *Figure 6—source data 2* for a complete list of predicted upstream regulators. n = 3 mice per genotype for (**A–C**). (**D**) A bi-clustering heatmap of the top 30 differentially expressed genes between wildtype and *Crtap*⁻/⁻ patellar tendons sorted by adjusted p-value and plotted according to $\log_2$ transformed expression values. The Wald test was used to generate p-values and $\log_2$ fold changes. See *Figure 6—source data 3* for a complete list of differentially regulated genes. (**E**) Significantly differentially expressed genes between patellar tendons from wildtype and *Crtap*⁻/⁻ mice were clustered by their gene ontology, and enrichment for gene ontology terms was tested using Fisher exact test. All gene ontology terms with an adjusted p-value<0.05 are plotted according to their $-\log_{10}$ adjusted p-value. (**F**) Select upstream regulators predicted as being activated (shown in green), inhibited (shown in red), or of unclear state (shown in gray) in *Crtap*⁻/⁻ compared to wildtype patellar tendons plotted according to $-\log_{10}$ adjusted p-value. See *Figure 6—source data 4* for a complete list of predicted upstream regulators. n = 2 mice per genotype for (**D–F**).

The online version of this article includes the following source data for figure 6:

**Source data 1.** List of differentially expressed genes between 1-month wildtype and *Crtap* knockout Achilles tendons.
**Source data 2.** Predicted upstream regulators driving differential gene expression between 1-month wildtype and *Crtap* knockout Achilles tendons.
**Source data 3.** List of differentially expressed genes between 1-month wildtype and *Crtap* knockout patellar tendons.
**Source data 4.** Predicted upstream regulators driving differential gene expression between 1-month wildtype and *Crtap* knockout patellar tendons.

To investigate whether loss of CRTAP was associated with increased ECM remodeling at skeletal maturity, we performed a series of immunohistochemistry experiments at 4 months-of-age (*Figure 8*). Interestingly, Herovici staining revealed an abundance of immature collagen in patellar tendons from *Crtap*⁻/⁻ mice compared to wildtype, as evidenced by the prominent blue staining (*Figure 8A,B*). We

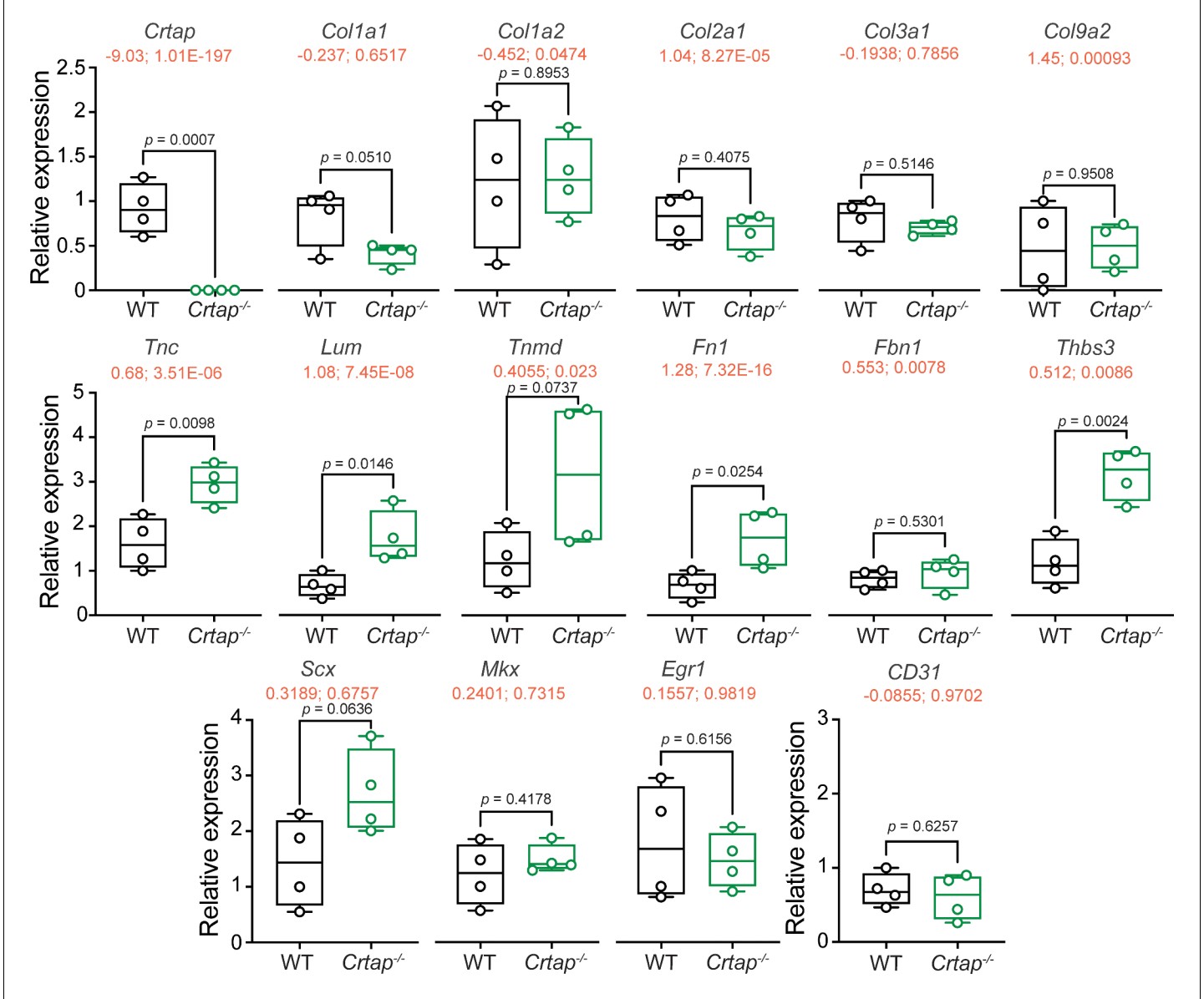

**Figure 7.** Tendon marker gene expression is altered in *Crtap*⁻/⁻ Achilles tendons at 1 month. Real-time quantitative PCR was performed to examine changes in the expression of *Crtap*, various fibrillar collagens (i.e., *Col1a1*, *Col1a2*, *Col2a1*, *Col3a1*, *Col9a2*), tendon makers (i.e. *Scx*, *Mkx*, *Egr1*, *Tnc*, *Lum*, *Tnmd*, *Fn1*, *Fbn1*), other targets from the RNA-seq analysis in **Figure 6** (i.e., *Thbs3*), and CD31 as a marker of vascularization. n = 4 mice per group. Data passed the Shapiro-Wilk test for normality, and groups were compared using one-way ANOVA with Tukey's post-hoc tests. Exact p-values are reported. The $\log_2$ fold change and adjusted p-value (shown as $\log_2$ fold change; adjusted p-value") from the RNA-seq experiment in **Figure 6** is indicated in red just below the gene name.

also observed increased α smooth muscle actin (αSMA; **Figure 8C,D**) and MMP2 (**Figure 8E,F**) together with elevated levels of phosphorylated nuclear factor kappa B (NFκB; **Figure 8G,H**), consistent with our RNA-seq data and our hypothesis that *Crtap*⁻/⁻ tendons exist in a state of increased remodeling/repair and inflammation.

## Loss of CRTAP leads to deficiencies in motor activity, coordination, and strength

Based on the observed defects in *Crtap*⁻/⁻ tendons, we performed a series of behavioral assays at 4 months-of-age to assess motor phenotypes. Using the open-field assay to quantify changes in spontaneous motor activity, we observed that *Crtap*⁻/⁻ mice displayed significant reductions in both

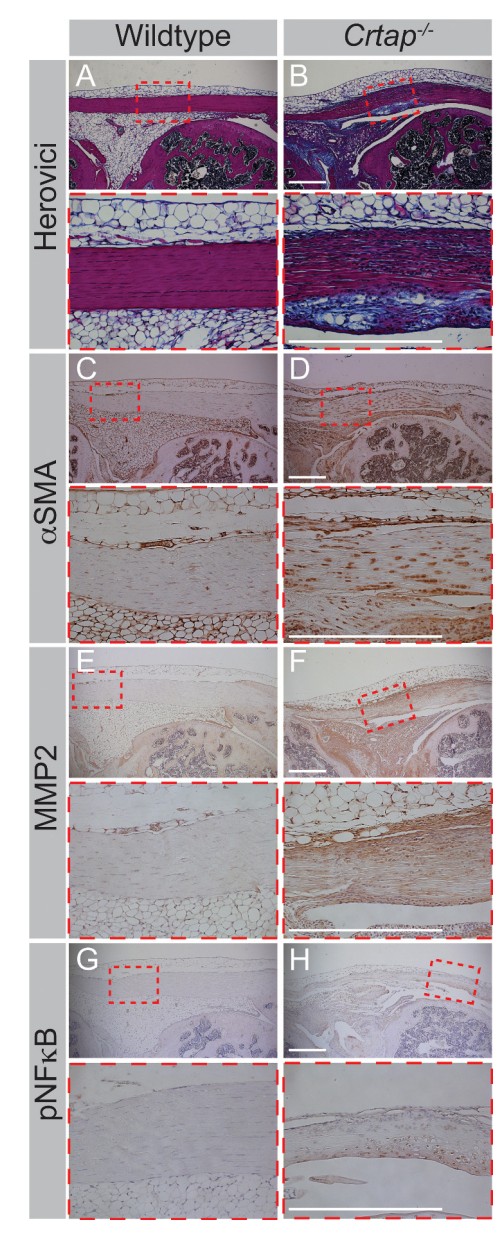

**Figure 8.** Four-month *Crtap*[-/-] patellar tendons exhibit increased staining for markers of fibrosis, increased ECM turnover, and inflammatory events. (**A–B**) Representative herovici-stained images of 4-month patellar tendons. (**C–D**) Representative αSMA IHC images of 4-month patellar tendons. (**E–F**) Representative MMP2 IHC images of 4-month patellar tendons. (**G–H**) Representative phosphorylated NFκB (pNFκB) IHC images of 4-month patellar tendons. For all micrographs, higher magnification images of the mid-tendon are illustrated. n = 3–5 mice per group. Scale bar is 0.5 mm.

horizontal and vertical activity compared to heterozygous and wildtype mice (*Figure 9A–B*). We next examined changes in motor coordination and endurance using the rotarod assay. While no genotype-dependent differences were observed during the learning phase of the assessment (Trials 1–5), *Crtap*[-/-] displayed a reduction in latency to fall for Trials 6, 9–10 compared to wildtype and Trials 6, 8–10 compared to heterozygous mice (*Figure 9C*). To confirm this observation, we evaluated the mice using the grid foot slip assay – an alternative metric for motor coordination. In this regard, we found that *Crtap*[-/-] mice exhibited a modest increase in forelimb and hindlimb foot slips compared to heterozygous and wildtype mice (*Figure 9D–E*). Taken together, these findings indicate that *Crtap*[-/-] mice have deficiencies in motor activity and coordination compared to controls.

We next evaluated strength in the *Crtap*[-/-] mice using the inverted grid and grip strength assays. Interestingly, we observed a decrease in the latency to fall during the inverted grid assay for *Crtap*[-/-] mice compared to heterozygous and wildtype controls (*Figure 9F*). Using a more quantitative metric, we examined these mice using the grip strength test and found that while wildtype and heterozygous mice could generate approximately 1.3 N of force, mice lacking CRTAP were weaker with a mean grip strength of 0.62 N (*Figure 9G*). Thus, *Crtap*[-/-] mice display significant reductions in strength together with perturbations in motor activity and coordination – behavioral changes that could be related in part to their tendon phenotype.

## Discussion

In this study, we examined the histological, ultrastructural, biochemical, and transcriptional characteristics of load-bearing tendons in the *Crtap*[-/-] mouse model of severe, recessive OI. We demonstrate that at 1 and 4 months-of-age, *Crtap*[-/-] have thinner, more cellular Achilles and patellar tendons with a reduction in the CD146[+]CD200[+] cell pool compared to heterozygous and wildtype mice. Importantly, we did observe only slight thinning with no increased cell density in Achilles or patellar tendons from *Crtap*[-/-] mice at P10, suggesting that the phenotype is mostly restricted to deficits in tendon maturation but not development. Examining collagen fibril organization, we found a marked alteration in collagen fibril alignment and the number of small and large fibrils in *Crtap*[-/-] mice compared to wildtype that varied in severity depending on the tendon

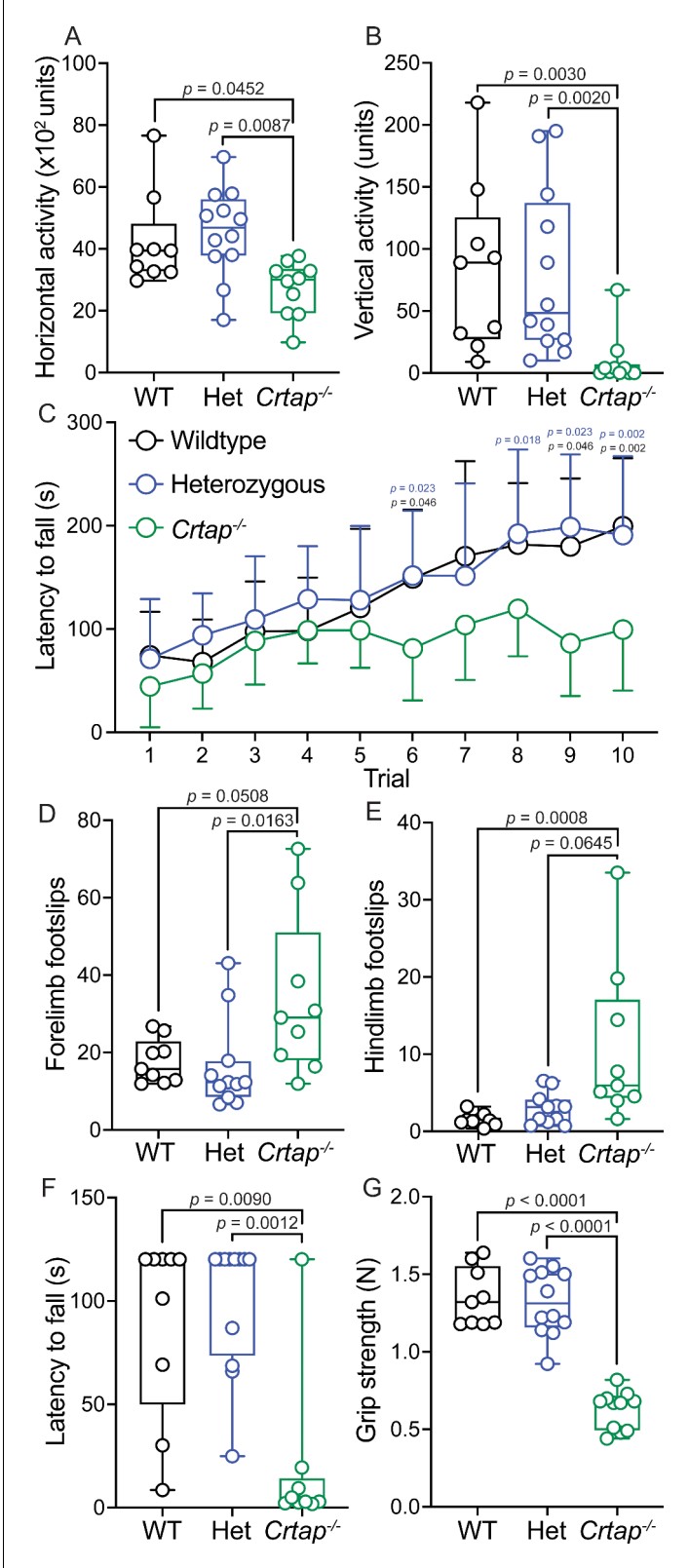

**Figure 9.** Motor activity and coordination are impaired in 4-month-old *Crtap*<sup>-/-</sup> mice. (**A–B**) Quantification of spontaneous motor activity including horizontal (**A**) and vertical (**B**) activity over a 30-min period using the open-field assay. Data are min-to-max box and whisker plots with individual points indicated. n = 9–12 mice per group. For (**A**), data passed the Shapiro-Wilk test for normality, and groups were compared using one-way ANOVA with
*Figure 9 continued on next page*

*Figure 9 continued*

Tukey's post-hoc tests. For (**B**), data failed the Shapiro-Wilk test for normality, and groups were compared using Kruskal-Wallis with Dunn's post-hoc tests. Exact p-values are reported for (**A–B**). (**C**) Quantification of motor activity, coordination, and endurance across 10 trials conducted over 2 days using an accelerating rotarod assay. Data are means ± S.D. n = 9–12 mice per group. Groups were compared using repeated-measures two-way ANOVA with Tukey's post-hoc tests. Exact p-values are reported, with black *p*-values being compared to wildtype, and blue p-values being compared to heterozygous mice. (**D–E**) Quantification of forelimb (**D**) and hindlimb (**E**) motor coordination using the grid foot slip assay. Data are min-to-max box and whisker plots with individual points indicated. n = 9–12 mice per group. Data failed the Shapiro-Wilk test for normality, and groups were compared using Kruskal-Wallis with Dunn's post-hoc tests. Exact p-values are reported. (**F**) Quantification of forelimb and hindlimb grip strength using the inverted grid assay conducted for 120 s. A reduction in latency to fall indicates reduced strength. Data are min-to-max box and whisker plots with individual points indicated. n = 8–9 mice per group. Data failed the Shapiro-Wilk test for normality, and groups were compared using Kruskal-Wallis with Dunn's post-hoc tests. Exact p-values are reported. (**G**) Quantification of forelimb grip strength in N of force measured over three trials and then averaged. Data are min-to-max box and whisker plots with individual points indicated. n = 9–12 mice per group. Data passed the Shapiro-Wilk test for normality, and groups were compared using one-way ANOVA with Tukey's post-hoc tests. Exact p-values are reported.

being examined. Consistent with our TEM findings, we found that 1-month Achilles tendons from *Crtap*$^{-/-}$ mice exhibited reduced ultimate load and linear stiffness compared to wildtype; however, the reduced structural properties could also be due to a reduction in cross-sectional area of the tendon at this age. Stable, HP cross-links were also elevated at 1- and 4-months in *Crtap*$^{-/-}$ mice compared to wildtype, which indicates an increase in telopeptide lysine hydroxylation and irreversible intermolecular cross-links (see below). RNA-seq analysis revealed alterations in extracellular matrix proteins, growth factor signaling, and metabolic pathways in *Crtap*$^{-/-}$ tendons compared to wildtype controls confirmed by qRT-PCR and IHC. Finally, we demonstrate that *Crtap*$^{-/-}$ mice exhibit motor impairments concomitant with reductions in grip strength – a phenomenon that may be related to the tendon pathology and possible deficits in other musculoskeletal tissues such as muscle.

Collagen ultrastructure and cross-linking have been well-documented in the bones of mouse models for both dominant and recessive OI; however, quantitative and histological characterization of tendons in these models has been less studied. In this regard, we demonstrate that *Crtap*$^{-/-}$ mice have thinner and weaker Achilles and patellar tendons at 1- and 4-months. Despite the reduction in size and extracellular matrix, tendons from these mice are hypercellular compared to wildtype and heterozygous animals – a phenotype observed in other tissues from *Crtap*$^{-/-}$ mice, including bone, lungs, and the glomerular compartment of the kidney (*Baldridge et al., 2010*; *Grafe et al., 2014*). Indeed, defects in ECM proteins such as type I collagen are correlated with changes in the proliferation and survival of surrounding cells (*Hynes, 2009*). Interestingly, the pathological changes observed in load-bearing tendons from *Crtap*$^{-/-}$ mice were accompanied by a significant reduction in CD146$^+$CD200$^+$ cells. Given that CD146 has been implicated as a marker for tendon progenitor-like cells (*Lee et al., 2015*), we speculate that the decrease in CD146 at 5-months may indicate that *Crtap*$^{-/-}$ tendons are in a state of chronic remodeling/repair (see below). While the alterations in tendon size and cellularity may result from alterations in the collagen extracellular matrix, it could also be associated with alterations in cellular signaling. In this regard, TGF-β signaling is upregulated in bones from *Crtap*$^{-/-}$ mice (*Grafe et al., 2014*), and elevated TGF-β signaling has been noted in mouse models with increased tendon cellularity and alterations in collagen fibril distribution (*Lim et al., 2017b*). Indeed, our RNA-seq analysis identified TGF-β signaling as an activated upstream regulator of the differential gene expression observed in the tendons from *Crtap*$^{-/-}$ mice. We also observed significant metabolic dysregulation in *Crtap*$^{-/-}$ tendons. Interestingly, mammalian target of rapamycin complex 1 (mTORC1) knockout mouse tendons have a similar histological phenotype to those of *Crtap*$^{-/-}$ mice (*Lim et al., 2017b*). These data are consistent with the principle that an altered ECM can alter cellular signaling, as previously shown for vascular tissue in Marfan syndrome and OI bone.

In addition to type I collagen, other minor collagens together with glycoproteins and SLRPs are present within tendons and regulate type I collagen self-assembly and are expressed most highly during development and repair. In the present study, we demonstrated that *Crtap* is expressed in load-bearing tendons and that its expression is completely ablated in tendons from *Crtap*$^{-/-}$ mice.

Other than a modest reduction in *Col1a1*, there was no significant difference in the expression of *Col3a1*, which is typically increased in the early stages of tendon injury and healing (*Howell et al., 2017*; *Juneja et al., 2013*; *Yang et al., 2013*). While *Col2a1* and *Col9a2* expression were upregulated in our RNA-seq data from *Crtap*[−/−] Achilles tendon, no differences were observed by qRT-PCR. Despite a lack of changes in the expression of many fibrillar collagens, we observed upregulation in the expression of *Scx* and various glycoproteins and SLRPs, including *Tnc*, *Lum*, *Tnmd*, *Fn1*, and *Thbs3*, in *Crtap*[−/−] mice. Many of these markers are upregulated during tendon injury and healing (*Yang et al., 2013*; *Snedeker and Foolen, 2017*), with *Tnmd* being a positive regulator of tenocyte proliferation (*Docheva et al., 2005*), elevated *Fn1* being a precursor for tendon ECM deposition (*Kadler et al., 2008*), and *Tnc* expression being elevated under conditions of increased mechanical load (*Chiquet et al., 2003*; *Chiquet-Ehrismann and Tucker, 2004*). Taken together, this would suggest that load-bearing *Crtap*[−/−] tendons are immature and stuck in a process of perpetual remodeling and repair compared to wildtype controls, reminiscent of the increased remodeling observed in *Crtap*[−/−] bones.

In the present study, we found an increased proportion of small and large collagen fibrils in the FDL, Achilles, and patellar tendons of *Crtap*[−/−] mice. These distributions varied in severity across the three tissues, with the most significant differences observed in the patellar tendon. *P3h1*[−/−] mice have been reported to display an increased proportion of small collagen fibrils in tail tendons (*Vranka et al., 2010*), indicating that loss of the 3-prolyl hydroxylase complex alters collagen fibrillogenesis. At the same time, the increased number of small fibers was more pronounced in tail tendons from *P3h1*[−/−] mice, and there was no evidence of an increased proportion of large fibers (*Vranka et al., 2010*) as observed for *Crtap*[−/−] mice in this study. Like the *P3h1*[−/−] mouse model, mice lacking CypB also exhibit a pronounced increase in the number of small collagen fibrils within tail tendons (*Terajima et al., 2016*). Together, this may indicate that despite forming a complex with P3h1 and CypB, loss of Crtap has distinct consequences on collagen fibrillogenesis. At the same time, it is important to note that tail tendons were used for collagen fibril distribution assessments in both *P3h1*[−/−] and *Ppib*[−/−] mice (*Terajima et al., 2016*; *Vranka et al., 2010*), whereas load-bearing tendons were examined in our study. The difference in tissue type examined might also explain why we observed unique alterations in collagen fibril distribution in both heterozygous and *Crtap*[−/−] mice. While the focus of the present study was on load-bearing tendon phenotypes and how these may relate to joint laxity and motor deficits seen in OI patients, future work examining positional tendons of the tail could also be insightful.

Type I procollagen molecules undergo post-translational modifications within the endoplasmic reticulum, including lysyl-hydroxylation and prolyl-hydroxylation, that are critical for proper collagen synthesis, transport, and stability. Specifically, telopeptide lysine hydroxylation results in mature lysyl-pyridinoline (LP) or hydroxylysyl-pyridinoline (HP) residues after lysyl oxidase oxidation, which as permanent, irreversible crosslinks play a role in regulating fibril growth and strength (*Eyre, 1987*; *Bateman et al., 2009*). Previous literature has demonstrated that loss of the 3-prolyl hydroxylase complex caused by loss of P3H1 (*P3h1*[−/−]) or CRTAP (*Crtap*[−/−]) prevents prolyl 3-hydroxylation of clade A (type I, II, and III) collagens and can lead to changes in lysine post-translational modifications due to loss of its chaperone function (*Hudson and Eyre, 2013*; *Morello et al., 2006*). In this study, we found that mature collagen cross-links (HP residues per collagen) are markedly increased in the FDL, Achilles, and patellar tendons of *Crtap*[−/−] mice relative to wildtype. This contrasts that seen in bones from *Crtap*[−/−] mice, where there is no change in HP residues per collagen compared to wildtype. Instead, an increase in the ratio of LP/HP is observed along with a complete lack of 3-hydroxylation of Pro 986 in the chains of type I collagen (*Baldridge et al., 2010*; *Grafe et al., 2014*). Interestingly, we observed increased cross-links in the Achilles, but not FDL or patellar tendons of 1-month-old heterozygous mice, indicating a mild haploinsufficient effect of CRTAP on this tendon biochemical property, thereby suggesting a rate-limited contribution of this complex in this function. Outside of the genotype-specific effects, we saw an age-dependent increase in HP residues per collagen in all genotypes. This observation is consistent with a study by Taga and colleagues that reported an increase in 3-hydroxyproline residues in rat tendon collagen (but not bone or skin) that plateaued at 3 months-of-age (*Taga et al., 2016*). Together with the TEM analyses, these data suggest that altered collagen cross-linking in tendons from *Crtap*[−/−] mice may adversely affect collagen fibril assembly.

In addition to skeletal deformities and frequent fractures, severe OI is associated with motor impairments, including gait abnormalities, chronic pain, and reduced muscle strength (*Arponen et al., 2014*; *Primorac et al., 2014*; *Garman et al., 2019*). In this study, we showed that *Crtap*⁻/⁻ mice exhibit reduced horizontal and vertical motor activity using the open field assay and reduced motor coordination and endurance using the rotarod and grid foot slip assays. We also observed a reduction in latency to fall on the inverted grid assay mirrored by a dramatic loss of grip strength compared to heterozygous and wildtype mice. These results are consistent with findings reported for the *Col1a1*^Jrt/+ mouse model of severe OI and Ehlers-Danlos syndrome (*Chen et al., 2014*). Specifically, Abdelaziz and colleagues found that *Col1a1*^Jrt/+ mice displayed reduced motor activity using the open field and running wheel assays – a phenotype they attributed to thermal hyperalgesia and mechanical allodynia in these mice (*Abdelaziz et al., 2015*). While the reduced vertical activity may indicate a pain or spinal phenotype, changes in motor coordination and grip strength in *Crtap*⁻/⁻ mice are likely more related to deficits in the muscle-tendon unit. Indeed, muscle dysfunction has been noted in OI patients and various preclinical mouse models (*Berman et al., 2020*; *Gremminger et al., 2019*; *Veilleux et al., 2017*). Unfortunately, given only a global *Crtap* knockout mouse line was used here, understanding the contribution of muscle independent of the tendon is complex and needs further study. Despite these limitations, our study describes the most comprehensive characterization of motor and strength deficits in a mouse model of severe, recessive OI to date, providing a useful set of functional outputs with which to evaluate future therapeutics or to interrogate the role of tendon and other musculoskeletal tissues in motor function.

Taken together, this study provides the first evidence for load-bearing tendon phenotypes in the *Crtap*⁻/⁻ mouse model of severe, recessive OI. We also provide compelling evidence for a strong motor activity and coordination phenotype in these mice. As the quality of life is so impacted in patients with OI, a more comprehensive evaluation of behavioral outcomes in future preclinical studies may provide important insights into the efficacy of therapeutic interventions.

## Materials and methods

**Key resources table**

| Reagent type (species) or resource | Designation | Source or reference | Identifiers | Additional information |
|---|---|---|---|---|
| Genetic reagent (*M. musculus*) | Crtap^tm1Brle | B. H. Lee Laboratory | DOI: 10.1016/j.cell. 2006.08.039 | Deposited at Jackson Labs (B6; 129S7-Crtap^tm1Brl e/J; Stock #: 018831) |
| Genetic reagent (*M. musculus*) | 129sv/EV | Dr. Allan Bradley | Baylor College of Medicine | Maintained in Lee Laboratory for many generations |
| Genetic reagent (*M. musculus*) | C57BL/6J | Jackson Laboratory | Stock #: 000664 RRID:IMSR_JAX:000664 | |
| Antibody | Anti-CD45- pacific blue (mouse monoclonal) | Invitrogen | Cat. #: MCD4528 RRID:AB_10373710 | FACS (1:100) Clone: 30-F11 |
| Antibody | Anti-CD31-eFluor 450 (mouse monoclonal) | Invitrogen | Cat. #: 48-0311-82 RRID:AB_10598807 | FACS (1:100) Clone: 390 |
| Antibody | Anti-CD146-PE-Cy7 (mouse monoclonal) | BioLegend | Cat. #: 134713 RRID:AB_2563108 | FACS (1:100) Clone: ME-9F1 |
| Antibody | Anti-CD200-APC (mouse monoclonal) | BioLegend | Cat. #: 123809 RRID:AB_10900996 | FACS (1:100) Clone: OX-90 |
| Antibody | Anti-αSMA (rabbit polyclonal) | Abcam | Cat #: ab5694 RRID:AB_2223021 | IHC (1:200) |
| Antibody | Anti-MMP2 (goal polyclonal) | R and D Systems | Cat #: AF1488 RRID:AB_2145989 | IHC (1:400) |

*Continued on next page*

*Continued*

| Reagent type (species) or resource | Designation | Source or reference | Identifiers | Additional information |
|---|---|---|---|---|
| Antibody | Anti-phospho-NFκB p65 (Ser536) (93H1) (rabbit monoclonal) | Cell Signaling | Cat #: 3033 RRID:AB_331284 | IHC (1:20) |
| Antibody | Biotin-SP-AffiniPure Donkey Anti-Rabbit IgG (H + L) | Jackson ImmunoResearch Labs | Cat #: 711-065-152 RRID:AB_2340593 | IHC (1:100, 1:400) |
| Antibody | Biotin-SP-AffiniPure Donkey Anti-Goat IgG (H + L) | Jackson ImmunoResearch Labs | Cat #: 705-065-147 RRID:AB_2340397 | IHC (1:500) |
| Antibody | Peroxidase-Streptavidin Slides were incubated with DAB substrate (Vector Laboratories, SK-4100) | Jackson ImmunoResearch Labs | Cat #: 016-030-084 RRID:AB_2337238 | IHC (1:100, 1:400, 1:500) |
| Antibody | DAB Substrate Kit (3,3'-diaminobenzidine) | Vector Laboratories | Cat #: SK-4100 RRID:AB_2336382 | |
| Other | Propidium iodide | Sigma-Aldrich | Cat. #: P4170-100MG | |
| Commercial assay or kit | RNeasy fibrous tissue mini kit | Qiagen | Cat. #: 74704 | |
| Commercial assay or kit | RNeasy micro kit | Qiagen | Cat. #: 74004 | |
| Commercial assay or kit | iScript cDNA synthesis kit | Bio-Rad | Cat. #: 1708890 | |
| Chemical compound, drug | LightCycler FastStart DNA Master SYBR Green I | Roche | Cat. #: 12239264001 | |
| Chemical compound, drug | Hexaammineruthenium (III) chloride | Sigma-Aldrich | Cat. #: 262005–5G | 0.7% w/v in phase-contrast μCT fixative, wash buffer, and post-fixative |
| Chemical compound, drug | Glutaraldehyde | Polysciences, Inc | Cat. #: 01909 | 2% v/v in phase-contrast μCT fixative |
| Chemical compound, drug | Cacodylic acid | Electron Microscopy Sciences | Cat. #: 12200 | 0.05 M in phase-contrast μCT fixative and wash buffer; 0.1 M for post-fixative |
| Chemical compound, drug | Osmium tetroxide | Electron Microscopy Sciences | Cat. #: 19190 | 1% w/v in phase-contrast μCT or TEM post-fixative |
| Software, algorithm | Fiji | ImageJ | https://imagej.net/Fiji RRID:SCR_002285 | Version 2.1.0/1.53 c |
| Software, algorithm | GraphPad Prism | GraphPad Software | https://graphpad.com RRID:SCR_002798 | Version 9.0.1 |
| Software, algorithm | Tri/3D BON | Ratoc System Engineering | https://www.ratoc.co.jp/ENG/3diryo.html | Version R.8.00.008-H-64 |
| Software, algorithm | FlowJo | BD | https://www.flowjo.com/solutions/flowjo/downloads RRID:SCR_008520 | Version 10.7 |

## Animals

*Crtap*$^{-/-}$ mice were generated as previously described (*Morello et al., 2006*) and maintained on a mixed C57BL/6J and 129Sv genetic background. Male mice were used for all experiments. All studies were performed with approval from the Institutional Animal Care and Use Committee (IACUC) at

Baylor College of Medicine. Mice were housed three to four mice to a cage in a pathogen-free environment with ad libitum access to food and water and under a 14 hr light/10 hr dark cycle.

## Histological analysis

Mice were euthanized and ankle and knee joints were dissected and fixed for 48 hr on a shaker at 4°C or room temperature in freshly prepared 4% paraformaldehyde (PFA) in 1 × phosphate buffered saline (PBS). Samples were decalcified at 4°C using 10% ethylenediaminetetraacetic acid (EDTA) in 1 × PBS for 10 days (with one change out at 5 days) before paraffin embedding using a standard protocol. Samples were sectioned at 6 μm and stained with hematoxylin and eosin (H and E) to visualize tendon structures. Herovici staining was performed on separate sections to visualize the ratio of mature collagen (red color) to more immature collagen and reticular fibers (blue color). Using H and E-stained tissue sections, cell number per tissue area as well as tissue area were determined using the Fiji release of ImageJ (*Schindelin et al., 2012*). For the 1-month and 4-month time points, two to threeregions of interest were selected in one plane-matched tissue section for each animal and used for quantification. For postnatal day 10 (P10), a single region of interest was selected across three different sections for each animal owing to the smaller size of the tissue at this earlier time point. For cell density, the number of cells was then divided by the tissue area to determine cell density.

## Fluorescence-activated cell sorting analysis of tendon progenitors

After dissection of patellar tendons from 5-month-old wildtype and *Crtap*[-/-] mice, tissues were cut into small pieces in 1 × PBS with 10% fetal bovine serum (FBS) and incubated with 500 μl of PBS + 10% FBS and 0.1% collagenase at 37°C for 3 hr. After digestion, cells were filtered with a 40 μm strainer, washed, resuspended in 1 × PBS at a concentration of $10^6$ cells/mL, and stained with CD45-pacific blue (clone: 30-F11), CD31-eFlour 450 (clone: 390), CD146-PE-Cy7 (clone ME-9F1) and CD200-APC (clone OX-90) (eBioscience). Propidium iodide was used for selecting viable cells. Cell analysis was performed using a LSRII Fortessa, and fluorescence-activated cell sorting (FACS) experiments were done using an AriaII cytometer (BD Biosciences, San Jose, CA). Data were analyzed with FlowJo software (TreeStar, OR).

## Phase-contrast μCT imaging and analysis

To quantify tendon and articular cartilage volume, knee joints were dissected from mice, stained with contrast agents, and scanned by phase-contrast μCT. The articular cartilage volume and surface were analyzed using TriBON software (RATOC, Tokyo, Japan) as previously described (*Nixon et al., 2018*; *Ruan et al., 2013a*; *Ruan et al., 2013b*; *Stone et al., 2019*). Using this technique, we quantified the patellar tendon volume by examining knee joints in transverse where the patellar tendon boundary was easily distinguished from the joint capsule. Tendon volume was assessed from its origin within the patella to its insertion at the tibia.

## Transmission electron microscopy analysis of collagen fibril size

Mouse ankle and knee joints were dissected and fixed in fresh 1.5% glutaraldehyde/1.5% PFA (Tousimis) with 0.05% tannic acid (Sigma) in 1 × PBS at 4°C overnight to preserve the native tension on relevant tendons. The next day, flexor digitorum longus (FDL), Achilles, and patellar tendons were dissected out in 1 × PBS, and placed back into fixative. Samples were then post-fixed in 1% osmium tetroxide ($OsO_4$), rinsed in Dulbecco's Modified Eagle Medium (DMEM), and dehydrated in a graded series of ethanol to 100%. Samples were rinsed in propylene oxide, infiltrated in Spurrs epoxy, and polymerized at 70°C overnight. TEM images were acquired using an FEI G20 TEM at multiple magnifications to visualize transverse and longitudinal sections of collagen fibrils. Collagen fibril diameter was measured using the Fiji release of ImageJ (*Schindelin et al., 2012*).

## Tendon biomechanical testing

Biomechanical tests were performed in tension on a universal testing machine (Instron 5848 Microtester) using a 100N load cell. The tests were performed under displacement control, at a rate of 0.1 N/s, until failure. Data was collected at 40 Hz, and stiffness and ultimate loads were calculated from the load-displacement curve. Tendons were fixed in the machine using a modification of the

clamping technique proposed by Probst and colleagues (*Probst et al., 2000*). The calcaneus was wedged into a conical slot created in a smooth plastic block, which was formed using a moldable plastic (InstaMorph) to ensure all edges were smooth and would not touch or damage the tendon. The origin of the tendon was affixed to the testing machine using a similar technique to that proposed by Probst. Testing was not performed in a water bath, but rather the tendons were kept moist with application of saline during each test, which lasted less than 60 s.

## Tendon collagen cross-linking analysis

Collagen hydroxylysyl-pyridinoline (HP) cross-links were quantified as previously described (*Eyre, 1987*; *Hudson et al., 2018*). In brief, tendons isolated from hindlimbs were hydrolyzed by 6M HCl for 24 hr at 108°C. Dried samples were then dissolved in 1% (v/v) n-heptafluorobutyric acid for quantitation of HP by reverse-phase HPLC with fluorescence monitoring.

## RNA-seq analysis

At 1 month-of-age, Achilles tendons were excised using scissors proximal to the calcaneal insertion and distal to the tendon-muscular-junction, whereas patellar tendon were removed via scalpel just proximal to the tibial insertion and distal to the patella. Tendons were not cleaned of their paratenon layers in either case. RNA extraction for both tissues was performed with the Fibrous Connective Tissue kit (Qiagen), with columns from the RNeasy Micro Kit (Qiagen) employed for the patellar tendon. Quality control, library preparation, sequencing, and differential gene expression analysis including gene ontology analysis was performed by GENEWIZ (South Plainfield, NJ). For the examination of predicted Upstream Regulators, we utilized the Ingenuity Pathway Analysis (IPA) platform (Qiagen) with an adjusted p-value of 0.05. The expression is shown as Base Means and read counts of genes were normalized per million transcripts (Transcripts Per Million; TPM).

## qRT-PCR analysis

At 1 month-of-age, Achilles tendons were excised from the calcaneus, immediately distal to the tendon-muscular-junction. RNA was extracted using the Fibrous Connective Tissue kit (Qiagen). qRT-PCR was performed on the LightCycler 96 System (Roche) using gene-specific primers and FastStart SYBR Green I (Roche) following cDNA synthesis with iScript (Bio-Rad). The sequences of primers used were as follows: *Crtap* fwd: 5'-GCTCTTTGACCAGAGTGACAGG-3'; *Crtap* rev: 5'-TCCTTC TGGAGCGTCGTCACAT-3'; *Col1a1* fwd: 5'-TTGGGGCAAGACAGTCATCGAAT −3'; *Col1a1* rev: 5'-TTGGGGTGGAGGGAGTTTACACGAA-3'; *Col1a2* fwd: 5'-AACCCATGAACATTCGCAC-3'; *Col1a2* rev: 5'-AACTCTCATTGGGATGGTCTACAC-3'; *Col2a1* fwd: 5'- GCTCATCCAGGGCTCCAATGATG TAG-3'; *Col2a1* rev: 5'- CGGGAGGTCTTCTGTGATCGGTA-3'; *Col3a1* fwd: 5'-GACCAAAAGGTGA TGCTGGACAG-3'; *Col3a1* rev: 5'-CAAGACCTCGTGCTCCAGTTAG-3'; *Col9a2* fwd: 5'-CAC-CAGGCATTGATGGCAAGGA-3'; *Col9a2* rev: 5'-AGGACCTCCTTTTGTTCCAGGC-3'; *Tnc* fwd: 5'-GAGACCTGACACGGAGTATGAG-3'; *Tnc* rev: 5'-CTCCAAGGTGATGCTGTTGTCTG-3'; *Lum* fwd: 5'-CCTTGGCATTAGTCGGTAGTGTCAGT-3'; *Lum* rev: 5'-CGATTTGGTTATTCCTCAGGTAAAG-3'; *Tnmd* fwd: 5'- CTTTACTAGGCTACTACCCATACCCCTACT-3'; *Tnmd* rev: 5'- ATATATTGGCTAACA-GAAGGTTAAGCGTTT-3'; *Fn1* fwd: 5'-CCCTATCTCTGATACCGTTGTCC-3'; *Fn1* rev: 5'-TGCCGCAACTACTGTGATTCGG-3'; *Fbn1* fwd: 5'-AGCCAGAACCTTCACATCATGGTACAAT-3'; *Fbn1* rev: 5'- AGCACCAAACAGACAACAGAAACCTA-3'; *Thbs3* fwd: 5'-GACCAGTGTGATGACGA TGCTG-3'; *Thbs3* rev: 5'-ACAGTTGTCGCAGGCATCACCA-3'; *Scx* fwd: 5'-AAGACGGCGA TTCGAAGTTAGAAG-3'; *Scx* rev: 5'-TCTCTCTGTTCATAGGCCCTGCTCATAG-3'; *Mkx* fwd: 5'-CAAGGACAACCTCAGCCTGAGA-3'; *Mkx* rev: 5'-CGGTGCTTGTAAAGCCACTGCT-3'; *Egr1* fwd: 5'-AGCGAACAACCCTATGAGCACC-3'; *Egr1* rev: 5'- ATGGGAGGCAACCGAGTCGTTT-3'; CD31 fwd: 5'- CCAAAGCCAGTAGCATCATGGTC-3'; CD31 rev: 5'- GGATGGTGAAGTTGGCTACAGG-3'.

## Immunohistochemistry

Immunohistochemistry was performed using anti-αSMA (Abcam, ab5694, 1:200), anti-MMP2 (R and D Systems, AF1488, 1:400), and anti-phospho-NFκB p65 (Ser536) (Cell Signaling (93H1), #3033, 1:20) antibodies. Briefly, slides of knee joints from 4 month wildtype and *Crtap*$^{-/-}$ mice were deparaffinized and treated with 3% hydrogen peroxide and proteinase K solution (20 mg/ml) for 10 min. Slides were blocked using 5% normal donkey serum and incubated for 1 hr. Primary antibodies were

diluted in blocking solution and incubated overnight at 4°C. Slides were washed with PBS and incubated in anti-rabbit-biotin (Jackson Immunoresearch, 711-065-152, 1:400 for αSMA or 1:100 for phospho-NFκB) or anti-goat-biotin (Jackson Immunoresearch, 705-065-147, 1:500 for MMP2) for 1 hr. Following PBS wash, slides were incubated in streptavidin-HRP (Jackson Immunoresearch, 016-030-084, 1:400–500 for αSMA and MMP2 or 1:100 for phosho-NFκB) for 30 min then washed with PBS. Slides were incubated with DAB substrate (Vector Laboratories, SK-4100) for approximately 2 min, counterstained with hematoxylin, then cleared and mounted with a coverslip.

## Open-field assessment of spontaneous motor activity

Open-field activity was measured using the VersaMax Animal Activity Monitoring System (AccuScan Instruments, Columbus, OH). On the day of assessment, mice were transferred to the test room and allowed to acclimate in their home cage for 30 min at 50 Lux of illumination with 60 dB of white noise. Mice were then placed individually into clear 40 cm ×40 cm × 30 cm chambers and allowed to move freely for 30 min. Locomotion parameters and zones were recorded using the VersaMax activity monitoring software. Chambers were cleaned with 30–50% ethanol to remove the scent of previously tested mice between each run.

## Rotarod analysis of motor coordination and endurance

On the day of assessment, mice were transferred to the test room and allowed to acclimate within their home cage for 30 min at 50 Lux of illumination with 60 dB of white noise. Mice were then placed on a rotarod (UGO Basile, Varese, Italy) set to accelerate from 5-to-40 RPM over 5 min. Five trials were performed per day for two consecutive days (trials 1–10) with a rest time of 5 min between trials. Latency to fall was recorded when the mouse fell from the rotating rod or went for two revolutions without regaining control. The rotarod was cleaned with 30–50% ethanol between mice to remove the scent of previously tested animals.

## Grid foot slip analysis of motor coordination

The grid foot slip assay consisted of a wire grid set atop a stand where the movement was recorded by a suspended digital camera. Mice were transferred to the test room on the day of assessment and allowed to acclimate within their home cage for 30 min at 50 Lux of illumination with 60 dB of white noise. Mice were then placed one at a time on the grid and allowed to move freely for 5 min. The observer sat 6–8 feet away at eye-level to the mouse and recorded forelimb and hindlimb foot slips using the ANY-maze video tracking software (Stoelting Co., Wood Dale, IL). After the test, mice were removed to their original home cage. Forelimb and hindlimb foot slips were normalized to the total distance traveled during the test.

## Inverted grid analysis of strength and endurance

On the day of assessment, mice were transferred to the test room and allowed to acclimate within their home cage for 30 min at 50 Lux of illumination with 60 dB of white noise. Mice were then placed in the middle of a wire grid, held approximately 18-in above a cushioned pad, and inverted. The latency to fall for each mouse was recorded. At the completion, mice were returned to their home cage.

## Grip strength analysis

Mice were transferred to the test room on the day of assessment and allowed to acclimate within their home cage for 30 min at 50 Lux of illumination with 60 dB of white noise. Each mouse was then lifted by its tail onto the bar of a digital grip strength meter (Columbus Instruments, Columbus, OH). Once both forepaws had gripped the bar, the mouse was gently pulled away from the meter by its tail at a constant speed until the forepaws were released. The grip (in N of force) was recorded and the procedure repeated twice for a total of three measurements, which were averaged for the final result.

## Statistical analysis

Determination of sample size was based on previous publications. Biological replicates were defined as an individual mouse for each experiment. Respective tendons from the left and right hindlimbs

were combined for TEM, collagen cross-linking, tendon progenitor, and RNA-seq analyses. Data are presented as means ± S.D. or min-to-max box and whisker plots with individual data points. For data whose residuals passed the Shapiro-Wilk test for normality, groups of two were compared using unpaired t-tests, and groups of three or more were compared using one-way ANOVA followed by Tukey's post-hoc tests. For data whose residuals did not have a normal distribution, groups of two were compared using a Mann-Whitney test, and groups of three or more were compared using Kruskal-Wallis followed by Dunn's post-hoc tests. For the Rotarod assay, where time was a variable, groups were compared using repeated measures two-way ANOVA followed by Tukey's post-hoc tests. For all tests reported above, statistical analysis was performed using Prism 9.0.1 (GraphPad Software, La Jolla, CA). For all tests, the exact p-value is reported, and a p-value of <0.05 was considered statistically significant.

## Additional information

### Funding

| Funder | Grant reference number | Author |
|---|---|---|
| Eunice Kennedy Shriver National Institute of Child Health and Human Development | HD024064 | Brendan H Lee |
| Baylor College of Medicine | | Brendan H Lee |
| National Institute of Arthritis and Musculoskeletal and Skin Diseases | AR373318 | David R Eyre |
| National Institute of Child Health and Human Development | | Brendan H Lee |

The funders had no role in study design, data collection and interpretation, or the decision to submit the work for publication.

### Author contributions

Matthew William Grol, Conceptualization, Formal analysis, Validation, Investigation, Visualization, Writing - original draft, Project administration, Writing - review and editing; Nele A Haelterman, Joohyun Lim, Formal analysis, Validation, Investigation, Writing - review and editing; Elda M Munivez, Kevin Lei, Cole D Kuzawa, Investigation; Marilyn Archer, Sara F Tufa, Formal analysis, Investigation; David M Hudson, Douglas R Keene, Catherine G Ambrose, David R Eyre, Formal analysis, Investigation, Writing - review and editing; Dongsu Park, Formal analysis, Investigation, Visualization, Writing - review and editing; Brendan H Lee, Conceptualization, Resources, Formal analysis, Supervision, Funding acquisition, Project administration, Writing - review and editing

### Author ORCIDs

Matthew William Grol (iD) https://orcid.org/0000-0001-6514-9066
Joohyun Lim (iD) https://orcid.org/0000-0001-9670-806X
Brendan H Lee (iD) https://orcid.org/0000-0001-8573-4211

### Ethics

Animal experimentation: This study was performed in strict accordance with the recommendations in the Guide for the Care and Use of Laboratory Animals of the National Institutes of Health. All of the animals were handled according to approved Institutional Animal Care and Use Committee (IACUC) protocols (#AN-1506) at Baylor College of Medicine.

### Decision letter and Author response

Decision letter https://doi.org/10.7554/eLife.63488.sa1
Author response https://doi.org/10.7554/eLife.63488.sa2

# Additional files

## Supplementary files
- Transparent reporting form
- Reporting standard 1. ARRIVE checklist.

## Data availability

All data generated or analyzed during this study are included in the manuscript and supporting files. Source data files have been provided for Figure 6 that include full lists of differentially expressed genes resulting from RNA-seq analysis of Achilles and patellar tendons from 1- month wild-type and *Crtap*⁻/⁻ mice. For each, a list of predicted upstream regulators identified using Ingenuity Pathway Analysis is also included.

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
