## [Decision Letter]

**Acceptance summary:**

The data presented provide evidence that loss of Crtap in mice, which is known to affect multiple connective tissues, results in a hypotrophic tendon phenotype. This Crtap null mouse model recapitulates many aspects of a rare form for Osteogenesis Imperfecta, a disease associated with increased low/no trauma fractures, increased risk of tendon ruptures and lax joint. As such the data presented in this paper partially explain the decreased muscle strength and the tendon phenotype seen in this complex human disease. The results of this study will be of interest to those studying pathologies affecting connective tissues in general, and in particular to those engaged in tendon biology research.

**Decision letter after peer review:**

Thank you for submitting your article "Tendon and motor phenotypes in the *Crtap^-/-^* mouse model of recessive Osteogenesis Imperfecta" for consideration by *eLife*. Your article has been reviewed by 2 peer reviewers, and the evaluation has been overseen by a Reviewing Editor and Mone Zaidi as the Senior Editor. The following individuals involved in review of your submission have agreed to reveal their identity: Andreas Traweger (Reviewer #1); Nathaniel Dyment (Reviewer #2).

The reviewers have discussed the reviews with one another and the Reviewing Editor has drafted this decision to help you prepare a revised submission.

Summary:

In this study, the authors conduct an in-depth examination of the tendon and ligament phenotype in *Crtap^-/-^* mice, a model of osteogenesis imperfecta. As is seen in patients, these mice present with deficient muscle strength, and you go on to show that this associated with a reduction in tendon size, reduced collagen fibril size and increased collagen cross-linking. In addition, you observed increased tendon cellularity and changes in gene expression in the tendons reflective of increased inflammation. There were many concerns raised in review, specifically about the model. First, this is a global mutant mouse, and it cannot be determined which aspects of the phenotype are due to deficits intrinsic to the tendon and which were secondary to the other musculoskeletal defects. Ideally the phenotype of a conditional knockout should be presented, but if such a model is not available this issue MUST be addressed in detail in the discussion. Second, there was universal concern among the reviewers about the lack of mechanical testing of the tendon tissue as is outlined in greater detail below. Lastly, there was a lively discussion among the reviewers about the perception that these results are preliminary and lack an in-depth investigation. Detailed comments are listed below.

Essential revisions:

1. Please clarify if *Crtap^-/-^* mice also display a phenotype in positional (less load-bearing) tendons, such as tail tendons? If a phenotype is mainly seen in load-bearing tendons, this would imply that the altered ECM triggers different responses due to mechanical load (as is outlined in the Discussion section, lines 408f).

2. The manuscript lacks an in-depth characterization of tendon and ligament tissue. Are there any differences in collagen fiber alignment, Collagen I to Collagen III ratio, etc.? Is Crtap actually expressed in tendons in wild type animals? You demonstrated a shift to smaller diameter collagen fibrils in these mice--is this consistent with previous results in bone? How about cross-links? Are the changes in tendon consistent with changes in bone?

3. You performed extensive functional (motor activity, coordination, and strength) and structural (TEM) assays and showed deficits in the *Crtap^-/-^* mice. However, material and structural properties would more directly show alterations in tendon function. As was noted, changes in motor/coordination/strength assays could be due in part to tendon phenotype but may also be due to muscle weakness and pain response. The decreased volume and collagen fibril size with increased cross-links suggests that biomechanical properties will likely be altered but these measures don't always correlate with mechanics. Tendon mechanical tests would be a great addition to the paper (e.g. Young's modulus, max. tensile stress).

4. Lines 258-59: Although *Crtap^-/-^* mice display a clear tendon phenotype, you did not investigate tendon development per se. The data merely implies that tendons do not mature properly, as no late embryonic and early postnatal stages have been investigated. The data provided do not allow for a specific conclusion to be made in this regard.

5. Lines 260ff: Based on the data provided it remains questionable if loss of Crtap in tendons is mechanistically responsible for the observed changes in cell density. Only very little data on cell characterization is provided, e.g. you do not investigate the expression of tendon-related markers (i.e IF for Tnmd, Scx, Mkx, Egr1) in situ. Further, a more detailed in vitro characterization of tendon stem and progenitor cells (TDSPCs) isolated from *Crtap^-/-^* tendon tissue would further substantiate this claim (e.g. proliferation rate, trilineage differentiation potential, etc.)

6. Although the result from RNASeq analysis are interesting, you do not verify any of their findings by independent means. At a minimum, a change in expression of the identified candidate genes should be demonstrated by qPCR and/or Western blot. Further, mechanistic insight for any of biological processes shown to be impacted by the loss of Ctarp-/- should be provided.

7. You provide results from several behavioral tests clearly demonstrating motor deficits. Surprisingly, none of test performed allow any conclusions to be drawn on the functional consequences of the tendon phenotype observed. Do the animals actually show any change in gait and/or endurance?

8. There is the claim that the behavioral changes observed can be related to the tendon phenotype. While this is a possibility, did you consider that abnormal muscle forces in part drive the tendon phenotype? Could it be that simply the reduced movement (due to pain) and loading of tendons are (mainly) causative for the observed tendon phenotype? This is actually suggested by the open field test results (Figure 6A/B).

9. Since collagen assembly and organization are disrupted in this model and with many OI models leading to increased bone remodeling, there is reason to be believe that OI tendons may be in a chronic healing response. The apparent increase in the number of CD31+ and CD45+ cells (Figure 2A-B), differential gene expression, and the extreme elongation of the patellar tendon that could be caused, in part, by microtears in the matrix, help to support this hypothesis. It would be interesting to stain for CD31 to see if there is an increased in vascularity or stain for matrix proteins that are elevated in healing or tendinopathic tissue (e.g., TNC, type III collagen, α SMA). A discussion of this mechanism is warranted.

10. Resident tendon progenitor populations are not well understood. While CD146 and CD200 may be markers of progenitors, related to my previous point, production of these proteins is likely increased in healing tissues and may not be specifically expressed in a progenitor population. We would caution labeling the CD146+/CD200+ cells as progenitors and the CD146-/CD200+ cells as immature cells without additional functional assays.

11. The patellar ligament should be referred to as the patellar tendon. Anatomically, the patellar tendon connects two bones but structurally and functionally it is a tendon. It functions in the extension mechanism of the knee and the ECM has less ground substance and is also more aligned than ligaments.

12. It was noted that articular cartilage volume was not altered but did not present quantitative data. We are surprised to not see changes in articular cartilage given prolyl 3-hydroxylation in type II collagen and the role that Crtap may have in this process. In addition, the alterations in subchondral bone and synovial thickening suggest that changes to articular cartilage may also occur.

[Editors' note: further revisions were suggested prior to acceptance, as described below.]

Thank you for resubmitting your work entitled "Tendon and motor phenotypes in the *Crtap^-/-^* mouse model of recessive Osteogenesis Imperfecta" for further consideration by *eLife*. Your revised article has been evaluated by Mone Zaidi (Senior Editor) and a Reviewing Editor.

Summary:

The data presented provide evidence that loss of Crtap in mice, which is known to affect multiple connective tissues, results in a hypotrophic tendon phenotype. This Crtap null mouse model recapitulates many aspects of a rare form for Osteogenesis Imperfecta, a disease associated with increased low/no trauma fractures, increased risk of tendon ruptures and lax joint. As such the data presented in this paper partially explain the decreased muscle strength and the tendon phenotype seen in this complex human disease. The results of this study will be of interest to those studying pathologies affecting connective tissues in general, and in particular to those engaged in tendon biology research.

The manuscript has been improved, but there are some remaining issues, particularly in relation to new data, that need to be addressed, as outlined below:

Essential Revisions:

1. The authors now include biomechanical data which was very much appreciated by the reviewers. However, the authors reported only structural properties (stiffness and max force) but not material properties (modulus and ultimate stress). As the authors also demonstrated that the Achilles tendon was thinner in the mutant mice (Figure 1), it's conceivable that the reduced structural properties could be due to a reduction in cross-sectional area of the tendon. This is only a minor limitation though given the altered ECM seen by TEM but should still be stated.

2. The authors now provide data, that a hypoblastic and hypercellular phenotype is not observed in early postnatal stages. This is an interesting and important finding. While the authors provide quantitative data on the cellularity, no quantitation on patellar tendon diameter is included. This would further underscore their findings.

3. The authors argue that load-bearing tendons in *Crtap^-/-^* mice undergo increased remodeling reminiscent of a chronic repair state. Unfortunately, they only provide RT-qPCR to support their claims. As it appears that lack of Crtap renders tendons more sensitive to overload-induced (even at physiological loads) catabolic events, it is important to demonstrate an increase in matrix-remodeling enzymes (e.g. MMPs, etc.), inflammatory events (e.g. expression of pro-inflammatory cytokines, NfkB activation), and fibrotic markers (e.g. α SMA as previously suggested in the prior review) on the protein level (IHC) driving the hypoblastic phenotype. Also, investigating Tgf-b signaling seems warranted. While RT-qPCR data is sometimes helpful the reviewers felt that for some of these issues, histopathological assessment such as IHC for MMPs and/or α-SMA to help validate the injury/healing response would be more informative.

4. Herovicis polychrome stain, which allows some differentiation between ColI/ColIII and also gives more information on the maturity of the tissue. It was felt that this would really strengthen the story being presented in this manuscript.

---

## [Author Response]

Essential revisions:1. Please clarify if Crtap^-/-^ mice also display a phenotype in positional (less load-bearing) tendons, such as tail tendons? If a phenotype is mainly seen in load-bearing tendons, this would imply that the altered ECM triggers different responses due to mechanical load (as is outlined in the Discussion section, lines 408f).

We thank the reviewers for this interesting question. As this manuscript's interest was in examining appendicular load-bearing tendons as they may relate to clinical manifestations in OI patients, such as joint laxity, we did not examine tail tendons in our mice. We have highlighted this fact now in the Discussion of our manuscript (Lines 381-388). In the lab, we have observed that *Crtap^-/-^* mice develop kinked tails with incomplete penetrance beginning at 1-2 months-of-age. This is a phenotype we are interested in pursuing in future work in addition to creating a conditional knockout model to address the specificity of the tendon phenotype.

2. The manuscript lacks an in-depth characterization of tendon and ligament tissue. Are there any differences in collagen fiber alignment, Collagen I to Collagen III ratio, etc.? Is Crtap actually expressed in tendons in wild type animals? You demonstrated a shift to smaller diameter collagen fibrils in these mice--is this consistent with previous results in bone? How about cross-links? Are the changes in tendon consistent with changes in bone?

We thank the reviewers for this point. We have added several new analyses and experiments to address these points.

Concerning collagen fibril alignment, we have added longitudinal TEM images to Figure 4 (see Figure 4M-R) that clearly show the irregular organization of collagen fibrils in *Crtap^-/-^* mice compared to wildtype (Lines 188-193).

We have also performed qRT-PCR to confirm our RNA-seq data and more closely examine alterations in tendon-specific gene expression in *Crtap^-/-^* tendons compared to wildtype. These results highlight that Crtap is expressed in tendon and is absent in tissue from *Crtap^-/-^* (Figure 7, Lines 261-276) – a finding that confirms the RNA-seq data in Figure 6.

As part of this new qRT-PCR analysis, we examined the expression of various fibrillar collagens, glycoproteins, SLRPs, tendon transcription factors, and CD31 (as a marker of vascularization). We demonstrate that while a modest reduction in *Col1a1* expression is observed with no alteration in *Col3a1* expression, there is increased expression of *Scx*, *Tnc*, *Lum*, *Tnmd*, *Fn1*, and *Thbs3*. As the reviewers pointed out below, this increase in tendon marker expression likely indicates that the tendon ECM is undergoing increased remodeling reminiscent of a chronic repair state. This new data is highlighted in Figure 7 and described on Lines 261-276. We discuss these results in the context of the existing literature on tendon maturation and repair on Lines 351-366 of the Discussion.

In the original submission, we discussed how the shift in collagen fibril diameters seen in FDL, Achilles, and patellar tendons from *Crtap^-/-^* mice is consistent with mouse models lacking the other two components of the 3-prolyl hydroxylase complex, namely P3h1 and CypB. At the same time, the reviewers bring up an important point regarding whether similarities exist between tendon and bone. To address this, we discuss that while we see increases in HP residues per collagen in *Crtap^-/-^* tendons, HP is not drastically changed in *Crtap^-/-^* bones; instead, it is a decrease in LP and resulting increase in HP/LP that is seen in bones from *Crtap^-/-^* mice. Moreover, there is a complete loss of 3-prolyl hydroxylation at Pro 986 in the chains of type I collagen (see Lines 397-403). Beyond collagen cross-linking, the reductions in ECM and increased cellularity are consistent with that seen in bones, lungs, and glomeruli of these mice (see Lines 328-333).

3. You performed extensive functional (motor activity, coordination, and strength) and structural (TEM) assays and showed deficits in the Crtap^-/-^ mice. However, material and structural properties would more directly show alterations in tendon function. As was noted, changes in motor/coordination/strength assays could be due in part to tendon phenotype but may also be due to muscle weakness and pain response. The decreased volume and collagen fibril size with increased cross-links suggests that biomechanical properties will likely be altered but these measures don't always correlate with mechanics. Tendon mechanical tests would be a great addition to the paper (e.g. Young's modulus, max. tensile stress).

We thank the reviewers for highlighting this point. We have added results from biomechanical testing of Achilles tendons from 1-month-old wildtype, heterozygous, and *Crtap^-/-^* mice in the revised manuscript. As is now shown in Figure 1T-U, we observed a significant decrease in the ultimate load and maximum stiffness in *Crtap^-/-^* tendons compared to the other two genotypes (see Lines 128-136).

4. Lines 258-59: Although Crtap^-/-^ mice display a clear tendon phenotype, you did not investigate tendon development per se. The data merely implies that tendons do not mature properly, as no late embryonic and early postnatal stages have been investigated. The data provided do not allow for a specific conclusion to be made in this regard.

We thank the reviewers for this important point. In the revised manuscript, we have added a new Figure (see Figure 2) examining the tendon phenotype in *Crtap^-/-^* mice at postnatal day 10 (P10). We still observed a slight thinning of the Achilles and patellar tendons at this earlier time point, but there was no increase in cell density in either tendon compared to wildtype. Based on this new data, we conclude that *Crtap* is required for tendon maturation and postnatal homeostasis but is not likely to be required for their initial formation and development (see Lines 137-142).

5. Lines 260ff: Based on the data provided it remains questionable if loss of Crtap in tendons is mechanistically responsible for the observed changes in cell density. Only very little data on cell characterization is provided, e.g. you do not investigate the expression of tendon-related markers (i.e IF for Tnmd, Scx, Mkx, Egr1) in situ. Further, a more detailed in vitro characterization of tendon stem and progenitor cells (TDSPCs) isolated from Crtap^-/-^ tendon tissue would further substantiate this claim (e.g. proliferation rate, trilineage differentiation potential, etc.)

We thank the reviewers for this important suggestion. As described above for Comment 2, we addressed the concerns regarding tendon-related marker expression by performing qRT-PCR for several of these markers (see Figure 7, Lines 261-276).

In terms of the tendon progenitor cells, the new qRT-PCR data does indeed suggest that changes in CD146^+^ cells could simply be indicative of a tendon in a chronic but pathologic state of repair. As such, we have de-emphasized the progenitor discussion in the revised manuscript and focused more on the repair hypothesis put forth by the reviewers. That being said, we agree that an interesting avenue for future research would be to examine the properties of the CD146^+^CD200^+^ cells identified in Figure 3 to verify if they at all represent a true progenitor population, and to examine how the decrease in CD146^+^CD200^+^ population in *Crtap^-/-^* mice may (or may not) contribute to the tendon phenotype. We now highlight this important future direction in the Discussion (see Lines 333-337).

6. Although the result from RNASeq analysis are interesting, you do not verify any of their findings by independent means. At a minimum, a change in expression of the identified candidate genes should be demonstrated by qPCR and/or Western blot. Further, mechanistic insight for any of biological processes shown to be impacted by the loss of Ctarp-/- should be provided.

We thank the reviewers for this important suggestion. As described above for Comment 2, we addressed the concerns regarding tendon-related marker expression by performing qRT-PCR for several of these markers (see Figure 7, Lines 261-276).

7. You provide results from several behavioral tests clearly demonstrating motor deficits. Surprisingly, none of test performed allow any conclusions to be drawn on the functional consequences of the tendon phenotype observed. Do the animals actually show any change in gait and/or endurance?

We thank the reviewers for this point. We have tried to clarify in the text what each behavioral test measures. We note that the rotarod assay is an established measure of both motor coordination and endurance. Unfortunately, we did not perform any gait analyses on these mice.

A significant issue in identifying the tendon's contribution to these behavioral metrics is the use of the whole-body *Crtap* knockout mouse model. In the Discussion of the revised manuscript, we acknowledge that muscle and other tissues may contribute to the phenotype in the *Crtap^-/-^* mice (see Lines 422-434). Future work employing a tissue-specific *Crtap* mouse model will be needed to interrogate this question.

8. There is the claim that the behavioral changes observed can be related to the tendon phenotype. While this is a possibility, did you consider that abnormal muscle forces in part drive the tendon phenotype? Could it be that simply the reduced movement (due to pain) and loading of tendons are (mainly) causative for the observed tendon phenotype? This is actually suggested by the open field test results (Figure 6A/B).

We thank the reviewers for this point and have added a section to the Discussion acknowledging that other tissues, namely muscle, could be contributing to behavioral metrics, including grid foot slip and grip strength (see Lines 422-434). We also discuss that to address such concerns, studies in a tissue-specific model will be required.

9. Since collagen assembly and organization are disrupted in this model and with many OI models leading to increased bone remodeling, there is reason to be believe that OI tendons may be in a chronic healing response. The apparent increase in the number of CD31+ and CD45+ cells (Figure 2A-B), differential gene expression, and the extreme elongation of the patellar tendon that could be caused, in part, by microtears in the matrix, help to support this hypothesis. It would be interesting to stain for CD31 to see if there is an increased in vascularity or stain for matrix proteins that are elevated in healing or tendinopathic tissue (e.g., TNC, type III collagen, α SMA). A discussion of this mechanism is warranted.

This is an excellent point made by the reviewers. We have performed qRT-PCR analysis of Achilles tendons from 1-month-old wildtype and *Crtap^-/-^* mice to address this. In addition to the upregulation of many tendon-specific genes (see the explanation for Comment 2), we examined the expression of CD31 at this timepoint and saw no difference between wildtype and *Crtap^-/-^* mice. In our tissue sections, we also did not observe any gross vascular infiltration into the tendon body. We agree with the reviewers that the upregulation of tendon markers combined with the changes seen in the ECM organization might indicate that the OI tendons are in a state of chronic healing. We highlight this point now in the Discussion (Lines 331-337, 351-366).

10. Resident tendon progenitor populations are not well understood. While CD146 and CD200 may be markers of progenitors, related to my previous point, production of these proteins is likely increased in healing tissues and may not be specifically expressed in a progenitor population. We would caution labeling the CD146+/CD200+ cells as progenitors and the CD146-/CD200+ cells as immature cells without additional functional assays.

We appreciate the reviewers' comments and acknowledge that the markers for tendon progenitors are not well-established. Indeed, based on the new qRT-PCR data, the changes in CD146^+^ cells could simply be indicative of a tendon in a chronic but pathologic state of repair. As such, we have de-emphasized the progenitor discussion in the revised manuscript and focused more on the repair hypothesis put forth by the reviewers.

11. The patellar ligament should be referred to as the patellar tendon. Anatomically, the patellar tendon connects two bones but structurally and functionally it is a tendon. It functions in the extension mechanism of the knee and the ECM has less ground substance and is also more aligned than ligaments.

We thank the reviewers for highlighting this error. We have now referred to it as the "patellar tendon" throughout the revised manuscript.

12. It was noted that articular cartilage volume was not altered but did not present quantitative data. We are surprised to not see changes in articular cartilage given prolyl 3-hydroxylation in type II collagen and the role that Crtap may have in this process. In addition, the alterations in subchondral bone and synovial thickening suggest that changes to articular cartilage may also occur.

We thank the reviewers for this point. In Figure 1R-S of the revised manuscript, we have included measurements of articular cartilage volume and surface at 4-months for wildtype, heterozygous, and *Crtap^-/-^* mice. In this analysis, we observed no drastic differences between groups (see Lines 121-122). Concerning genetic mouse models of early-onset osteoarthritis, this finding is not surprising as articular cartilage degeneration often occurs at least 6-months postnatally in the absence of a stressor (i.e., trauma).

[Editors' note: further revisions were suggested prior to acceptance, as described below.]

The manuscript has been improved, but there are some remaining issues, particularly in relation to new data, that need to be addressed, as outlined below:Essential Revisions:1. The authors now include biomechanical data which was very much appreciated by the reviewers. However, the authors reported only structural properties (stiffness and max force) but not material properties (modulus and ultimate stress). As the authors also demonstrated that the Achilles tendon was thinner in the mutant mice (Figure 1), it's conceivable that the reduced structural properties could be due to a reduction in cross-sectional area of the tendon. This is only a minor limitation though given the altered ECM seen by TEM but should still be stated.

We thank the reviewers for this observation. We have now noted this minor (but important) limitation in the Discussion of our manuscript (Lines 295-298).

2. The authors now provide data, that a hypoblastic and hypercellular phenotype is not observed in early postnatal stages. This is an interesting and important finding. While the authors provide quantitative data on the cellularity, no quantitation on patellar tendon diameter is included. This would further underscore their findings.

We thank the reviewers for this point. While we did not have P10 samples harvested for phase-contrast µCT analysis, we examined differences in tissue diameter by quantifying average tissue area mid-tendon for Achilles and patellar tendons from wildtype and *Crtap^-/-^* mice at this age. This data (Figure 2F, Lines 126-128) shows that there was only a slight decrease in mid-tendon area for the Achilles tendon (though not significant), and no difference in mid-tendon area for the patellar tendon between genotypes at P10.

3. The authors argue that load-bearing tendons in Crtap^-/-^ mice undergo increased remodeling reminiscent of a chronic repair state. Unfortunately, they only provide RT-qPCR to support their claims. As it appears that lack of Crtap renders tendons more sensitive to overload-induced (even at physiological loads) catabolic events, it is important to demonstrate an increase in matrix-remodeling enzymes (e.g. MMPs, etc.), inflammatory events (e.g. expression of pro-inflammatory cytokines, NfkB activation), and fibrotic markers (e.g. α SMA as previously suggested in the prior review) on the protein level (IHC) driving the hypoblastic phenotype. Also, investigating Tgf-b signaling seems warranted. While RT-qPCR data is sometimes helpful the reviewers felt that for some of these issues, histopathological assessment such as IHC for MMPs and/or α-SMA to help validate the injury/healing response would be more informative.

We thank the reviewers for highlighting this point and agree that IHC analysis for differences between matrix-remodeling enzymes, inflammatory events, and fibrotic markers would better support out conclusions. We have now performed IHCs for αSMA, *MMP2*, and phospho-NFκB on patellar tendon sections from 4-month-old wildtype and *Crtap^-/-^* mice (Figure 8). Consistent with our RNAseq analysis (Figure 6) and qPCR data (Figure 7), we observed increased levels of all three markers in *Crtap^-/-^* mice compared to controls (Figure 8C-H, Lines 252-260).

4. Herovicis polychrome stain, which allows some differentiation between ColI/ColIII and also gives more information on the maturity of the tissue. It was felt that this would really strengthen the story being presented in this manuscript.

As requested by the reviewers, we performed Herovici staining on patellar tendon sections from 4-month-old wildtype and *Crtap^-/-^* mice (Figure 8). We now show that there is an increase in immature collagen (indicated by blue staining) in *Crtap^-/-^* patellar tendons compared to wildtype (Figure 8A-B, Lines 254-256).